



# Earthquake-induced landslides in Haiti: seismotectonic and climatic influences, size-frequency relationships

Hans-Balder Havenith[1], Kelly Guerrier[2], Romy Schlögel[1,3], Anne-Sophie Mreyen[4], Sophia Ulysse[2,5], Anika Braun[6], Karl-Henry Victor[2], Newdeskarl Saint-Fleur[2], Lena Cauchie[1], Dominique Boisson[2], Claude Prépetit[5]

[1]University of Liege, Department of Geology, Georisk and Environment, Liege, 4000, Belgium

[2]Université d'Etat d'Haïti, Faculté des Sciences, LMI-CARIBACT, URGéo, Port-au-Prince, 10 Impasse Ambroise, Haiti

[3]Centre Spatial de Liège, Liege, 4000, Belgium

[4]University of Liege, Department of Urban & Environmental Engineering, Applied Geophysics, Liege, 4000, Belgium

[5]Unité Technique de Sismologie, Bureau des Mines et de l'Energie, Port-au-Prince, Delmas 31, Haiti

[6]TU Berlin, Faculty VI Planning Building, Environment Department of Engineering Geology, Berlin, 1587, Germany

Correspondence to: Hans-Balder Havenith (hb.havenith@uliege.be)



**Abstract.** First analyses of landslide distribution and triggering factors are presented for the region
affected by the August, 14, 2021, Nippes, earthquake (Mw=7.2) in Haiti. Landslide mapping was mainly
carried out by comparing pre- and post-event remote imagery (~0.5 -1-m resolution) available on Google
Earth Pro® and Sentinel-2 (10-m resolution) satellite images. The first covered about 50% of the affected
region (for post-event imagery and before completion of the map in January 2022), the latter were
selected to cover the entire potentially affected zone. On the basis of the completed landslide inventory,
comparisons are made with catalogues compiled by others both for the August 2021 and the January
2010 seismic events, including one open inventory (by the United States Geological Survey) that was
also used for further statistical analyses. Additionally, we studied the pre-2021 earthquake slope stability
conditions. These comparisons show that the total number of landslides mapped for the 2021 earthquake
(7091) is smaller than the one observed by two other research teams for the 2010 event (e.g., 23,567, for
the open inventory). However, these fewer landslides triggered in 2021 cover much wider areas of slopes
(>80 km$^2$) than those induced by the 2010 event (~25 km$^2$ – considering the open inventory). A simple
statistical analysis indicates that the lower number of 2021-landslides can be explained by the 'under-
mapping' of smallest landslides triggered in 2021, partly due to the lower resolution imagery available
for most of the areas affected by the recent earthquake; this is also confirmed by an inventory
completeness analysis based on size-frequency statistics. The much larger total area of landslides
triggered in 2021, compared to the 2010 earthquake, can be related to different physical reasons: a) the
larger earthquake magnitude in 2021; b) the more central location of the fault segment that ruptured in
2021 with respect to coastal zones; c) and possible climatic preconditioning of slope stability in the 2021-
affected area. These observations are supported by (1) a new pre-2021 earthquake landslide map, (2)
rainfall distribution maps presented for different periods (including October 2016 - when Hurricane
Matthew had crossed the western part of Haiti), covering both the 2010 and 2021 affected zones, as well
as (3) shaking intensity prediction maps.
**1    Introduction**
This paper presents a first overview of landslide hazards induced by the August 14, 2021, Nippes (Haiti)
earthquake. The epicenter (18.434° N / 73.482° W, hypocentral depth of 10 km) of this event is located



in the western part of the southern Haitian peninsula (see Unites States Geological Survey, USGS,
Earthquake Hazard Program page, earthquake.usgs.gov, presenting first information on the 2021 M 7.2
- Nippes, Haiti, event). Similar to the January 12, 2010, earthquake, the epicenter is located near the
surface expression of the Enriquillo-Plantain-Garden Fault (EPGF) that crosses the peninsula from East
to West, marking one of the highest seismic hazard zones of the island (see location of the epicenters on
the seismic hazard map completed by Frankel et al. in 2011, as well as on the topographic map shown in
Fig. 1).
For the 2010 event, Calais et al. (2010) and Symithe et al. (2013) showed that this earthquake was caused
by the oblique rupture of a formerly unknown fault dipping towards the North and located immediately
in the North of the EPGF. Data provided by the earthquake.usgs.gov webpage (considering the provided
moment tensor solution; see also Okuwaki and Fan, 2022) indicate that the situation could be similar for
the 2021 event, with a ruptured fault segment dipping towards the North, and mostly located in the North
of the EPGF. Thus, also the recently ruptured fault segment would not belong to the EPGF (which is
essentially a left-lateral strike-slip fault). It could be related to an adjacent blind fault segment with
oblique slip character (left-lateral strike-slip combined with reverse movement) according to the
information available on earthquake.usgs.gov. As by now there is no clear answer to this question, below
we will use the term of the 'EPGF zone' that includes the main strike-slip fault and annexed oblique slip
fault segments (the two that are now known, i.e., the one ruptured in 2010 and the one that produced the
last earthquake) to denominate the tectonic structure that produced those two events.
Even though the magnitude of the 2021 earthquake is slightly larger than the one of 2010 (Mw=7.2 and
Mw=7.0, respectively, see information on the earthquake.usgs.gov webpage and by Stein et al., 2021),
the recent event was far less catastrophic as it hit a less populated area compared to the 2010 earthquake
that occurred just near the western entrance of the capital of Haiti, Port-au-Prince. The 2021 earthquake
accounts for about 2250 fatalities (2/3 of which occurred in the provincial city of Les Cayes, located in
Fig. 1), while the 2010 death toll is up to 300,000. However, it quickly became clear that the last event
caused widespread slope failures that could be more intense than in 2010. Therefore, members of our
research teams completed some ground control during field visits along segments of important roads hit
by rock falls near the epicentral region. Additionally, we mapped all landslides visible on high-resolution
(<=1 m) satellite imagery available on Google Earth Pro® (GEPro, zooming to a scale of about 1/2500),



starting from August 28, 2021 (the first post-seismic high-resolution imagery available on GEPro), until
the end of October 2021 (comparing also with available pre-2021 imagery). This study was
complemented by landslide identification on Sentinel-2A and 2B products (10-m spatial resolution)
sensed for the period from August 14, 2021, until the end of September 2021 (especially for areas not
covered by higher resolution imagery on GEPro). This way, we could map landslides over the whole
area potentially hit by the 2021 event by using this imagery, as it will be explained in section 2.
After completion of the 2021 landslide inventory, statistical characteristics of the latter were compared
with equivalent results obtained for the 2010 USGS catalogue by Harp et al. (2016); some statistical data
are also compared with those of one other inventory completed by Martinez et al. (2021, USGS Open
File report) for the 2021 event and of two additional catalogues compiled for the 2010 event (by Gorum
et al., 2013 and Xu et al., 2014). A size-frequency analysis was carried out to assess the inventory
completeness (using the method proposed by Malamud et al., 2004) for our 2021 and the USGS open
2010 landslide catalogues.
We also mapped landslides existing before the 2021 earthquake by using high-resolution (<=1 m)
imagery available on GEPro starting from October 2014 until the end of 2017, to study some
preconditioning of slope instability that was induced in 2021. In particular, it is known that the region is
regularly affected by hurricanes – the last catastrophic hurricane had impacted the target area in October
2016: 'Matthew' or 'Mathieu' in French (see track roughly outlined in Fig. 1b). Also, just two days after
the main shock, on August 16, another Hurricane, 'Grace', hit the area and hampered help convoys to
reach the areas most impacted by the earthquake. Right after this event, it was not immediately clear if
'Grace' had contributed to landslide activity or not; this question will be analyzed in the following
sections by comparing landslide distributions with monthly precipitation maps produced by the 'Global
Precipitation Measurement' (GPM) Mission (NASA) for different periods.



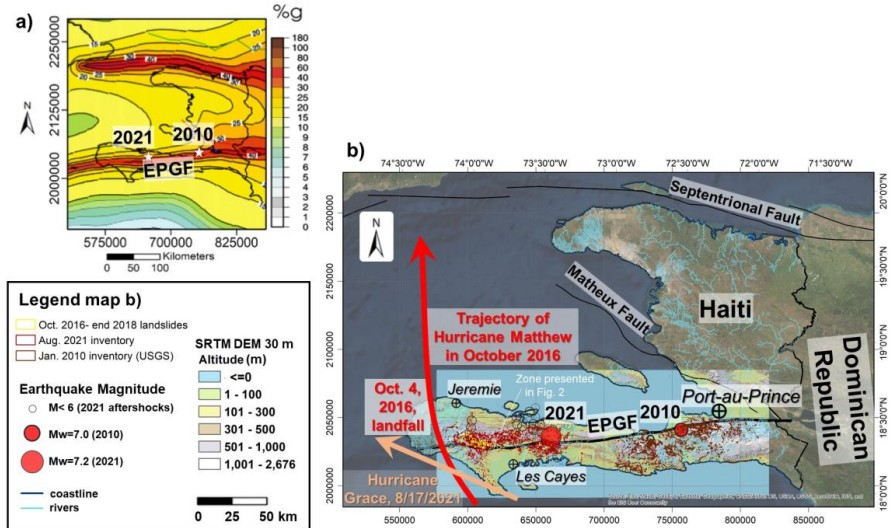


**Figure 1: Location of the study region in Haiti. a) Seismic hazard map of Haiti (modified from Frankel et al.,**
**2011) with location of the January 12, 2010, and August 14, 2021, main epicenters. b) Topographic map of**
**Haiti (by © ESRI 'with study region highlighted. See also location of the 2010 and 2021 epicenters and of the**
**cities with the largest number of victims caused by each of the events (the capital Port-au-Prince, hit in 2010**
**and the provincial city of Les Cayes with largest effects in 2021), the outline of the main active fault system in**
**the southern part of Haiti, the Enriquillo-Plantain-Garden-Fault (EPGF), and indication of the approximate**
**trajectory of Hurricane Matthew in October 2016. Landslides mapped by Harp et al. (2016) are shown by**
**brownish polygons (near the 2010 M=7 epicenter), and recently mapped landslides triggered in August 2021**
**are outlined in dark red (mainly in the West and South of the 2021 epicenter). Other digital outlines, including**
**faults and coastline, were provided by the Centre National de l'Information Géo-Spatiale (CNIGS) of Haiti.**
**See also location of the zone presented in Fig. 2.**

Finally, we also present a comparison of the 2010 and 2021 landslide distributions with respect to Arias
Intensity (Ia, see Arias, 1970) prediction maps, computed for each event by using the attenuation law
proposed by Keefer and Wilson (1989).


**2    Methodological aspects of landslide and seismic trigger factor mapping**
**2.1 Landslide mapping**
Right after the main shock that hit Haiti on August 14, 2021(precisely at 12:29:08 UTC, about 8:30 am
local time), it became clear that many landslides were triggered by this earthquake. Within a few hours
after the main shock, there were reports about rock falls cutting the main road RN7 connecting the large
provincial cities of Les Cayes in the South and Jeremy in the North. Therefore, members of our local
research teams checked the situation to support local administration with cleaning the roads. Photographs
of rock falls in the central part of the target area are shown in Fig. 2 (those shown below all occurred in
limestone rocks), together with the locations of the affected sites on a map.

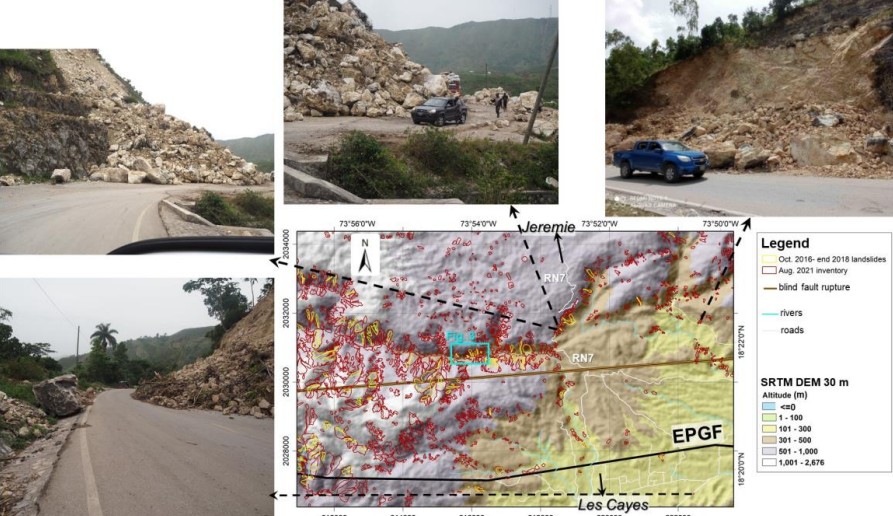


**Figure 2: Examples of landslides photographed in the field, especially along the national road RN7, connecting**
**the two provincial cities of Les Cayes in the South and Jeremy in the North. This map (located in the map of**
**Fig. 1) also shows the different ground failure effects mapped before (yellow polygons) and after the**
**earthquake (dark red polygons). See blue rectangle marking the outline of the view extent shown in Fig. 3,**
**presenting a more detailed analysis of the co- versus pre-earthquake landslide activity. Other digital outlines,**
**including rivers (light blue lines) and roads (white lines), were provided by the Centre National de**
**l'Information Géo-Spatiale (CNIGS), Haiti.**



These rock falls were typically not very large (with a volume of generally less than 20,000 m$^3$), but there
were many of them and in some cases, it took several days before the street could be reopened. For that
reason, our research groups, who have been working for several years on earthquake risk mitigation and
communication in some areas hit by the 2021 event, started to detect and map all landslides caused by
the earthquake. During field visits in August 2021, just after the main shock, our teams could confirm
that this earthquake had triggered more extensive slope failures (covering wider surface areas) than the
previous M=7.0 event in January 2010.
For the latter, several research groups had mapped co-seismic landslides (Gorum et al., 2013; Xu et al.,
2014; Terrier et al., 2014, for urban areas and Harp et al., 2016). In the following section, a few statistical
characteristics of the new 2021 landslide inventory are compared with those of the two first catalogues;
a more detailed comparison is completed with the Harp et al. (2016) inventory, freely available online.
For the landslide mapping over the whole potentially affected area we have used medium-resolution
imagery available from the Copernicus Open-Access Hub: Sentinel-2, with 10-m spatial resolution bands
B2 (490 nm), B3 (560 nm), B4 (665 nm) and B8 (842 nm) collected for 8 different dates, every five to
six days, between August 14, 2021 (the first one was available about two hours after the main shock),
and the end of September 2021. Analyzing all images was necessary due to the extensive (but spatially
variable) cloud cover present on each image. Considering that only this medium-resolution imagery was
freely available in the beginning, we had decided to outline coalescent debris slides and flows marked
by a main common part by one single coherent landslide polygon; this aspect will have to be taken into
consideration when interpreting the landslide size-frequency statistics presented in the next section. As
introduced above, during the following months, higher resolution (0.5-1 m) imagery became available
on GEPro, which was then used to refine the initial landslide outlines, and to map smaller slope failures.
At the end of October 2021 (and still in the beginning of 2022), about 50% of the potentially affected
area was covered by higher resolution imagery (especially for the eastern part of this area). However, to
maintain a coherence within the catalogue, the main rule to map coalescent slides with a major common
part (and presenting the same aspect and type of failure) as one single mass movement was still respected.
Examples of landslides mapped by applying this technique are presented in Fig. 3. Areas covered both
by Sentinel-2 imagery and higher resolution images were also used to refine landslide mapping based on
the first type of imagery, as by the end of 2021, only this one with 10-m resolution was available for the





western zone (compare Fig. 3c and 3d). As a comparison with pre-event imagery was necessary in many
cases to be sure that only 'co-seismic' (or nearly co-seismic – see explanation below) slope failures had
been mapped, the whole area was screened by using high resolution (0.5-1 m) imagery available on
GEPro for the period between 2014 and August 2021. A pre-earthquake image (of November 28, 2014)
is shown in Fig. 3a, highlighting the contrast between the densely vegetated slopes present in the target
region and the extensive denudation that occurred during the earthquake of August 2021 (see images
shown in Figs. 3c and 3d). However, we could also observe by comparing multiple images available for
the pre-event period that some denudation had already appeared for smaller zones before 2021. Zones
marked by narrow debris slides and flows could be outlined especially on images available for the period
between October 10, 2016 and the end of 2017. Fig. 3b presents an image of October 12, 2016 that shows
the 'freshest' type of denudation since 2014 (see yellow polygons outlining such denudation zones), some
of which disappeared after two years, due to revegetation of the slopes (rapid revegetation can be
observed as the whole area is located in tropical regions). This image and others available for the same
period were added to GEPro after Hurricane Matthew had impacted, in early October 2016, the same
area as the one hit by the 2021 earthquake. The consequences of this 'double' impact on the target region
will be analyzed in the sections 3 and 4 on the basis of precipitation distribution maps.
Actually, Haiti is quite often (at least once per year) crossed by hurricanes or tropical storms, some of
which can trigger slope failures over wide areas. One such tropical storm that later developed into the
hurricane called 'Grace' had also crossed southern Haiti, just two to three days after the August 14, 2021,
main shock. We introduce this fact here in the methodological part as it had two consequences for the
landslide mapping: first, right after the earthquake wide areas were covered by clouds during several
days (some higher mountain parts even for weeks); thus, multiple satellite images of different dates (both
Sentinel-2 and higher resolution imagery on GEPro had to be inspected to map landslides over the whole
area. Second, we had to consider that 'Grace' might also have induced slope failures and that landslides
mapped by using post-hurricane imagery were not all seismically triggered, or were at least enlarged by
the effects of 'Grace'. Therefore, by comparing the post-seismic, August 14, Sentinel-2 image (collected
before the Hurricane Grace event) with the one of August 29, 2021 (post-seismic and post-hurricane),
we checked if additional or enlarged slope failures had appeared on the latter. An example of such a
comparison is presented in Fig. 4, where red arrows point to zones marked either by new or by larger
slope failures on the Sentinel-2 image of August 29, 2021, which were thus most likely caused by rainfall
during the Grace climatic event (disregarding here the possible additional influence of aftershocks
occurring during the same time in the region, which will be discussed below). Unfortunately, due to the
extensive cloud cover in mid-August 2021, such a comparison could only be completed for about 10%
of the seismically impacted area. For those cloud-free zones, we estimate that Grace had induced a
widening of slope denudation of about 10-15% compared to the purely seismically triggered slope
failures. As most images were available after the Grace event, the total number of 7091 landslides
mapped for the period between August 14 and the end of October 2021 inevitably also includes
precipitation-induced or reactivated slope failures (thus increasing the initial total number and area of
co-seismic landslides by about 10-15%). This aspect has also to be taken into consideration when the
landslide impact of the 2021 earthquake is compared with the one of 2010 (as analyzed in the next
section).

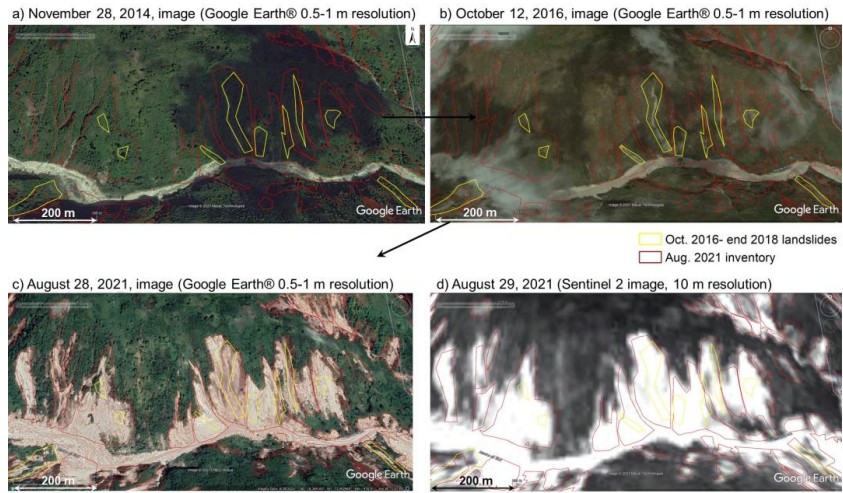


**Figure 3: Outlines of landslides mapped between October 2016 – November 2017 (in yellow; most by using**
**October 10-12, 2016, images, directly after Hurricane Matthew) and after August 14, 2021 earthquake (dark**
**red polygons), plotted on © Google Earth Pro, (a) high-resolution image available in GEPro for November 28,**
**2014, on (b) high-resolution image available in GEPro for October 12, 2016, on (c) high-resolution image**
**available in GEPro for August 28, 2021, and on (d) 10-m resolution Sentinel-2 image of August 29, 2021**
**(appearing in Black-White only in GEPro – in color in our GIS database).**

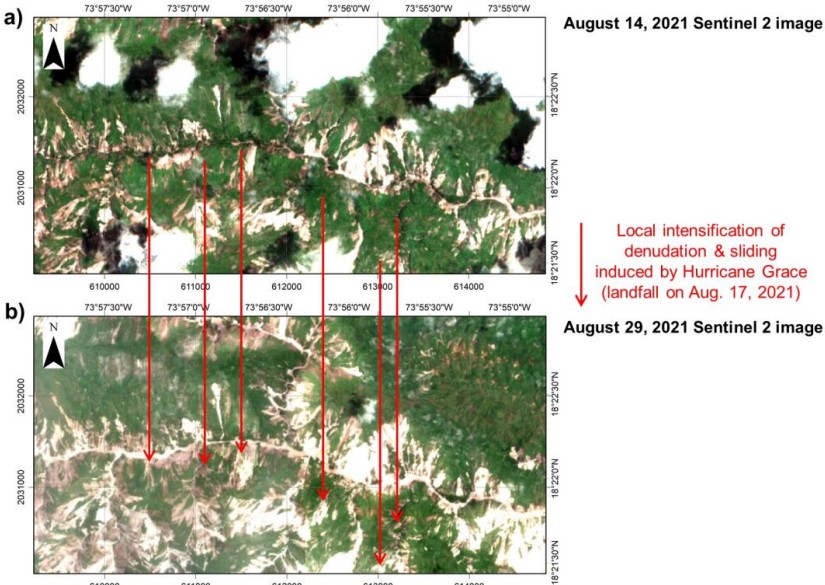


**Figure 4: Comparison between Sentinel-2 images (10-m resolution) for the same area obtained for (a) August**
**14 (about 2h after the main shock) and for (b) August 28, 2021 (12 days after impact by Hurricane Grace that**
**crossed the region on August 16, 2021). Red arrows point to zones where an intensification of denudation and**
**sliding can be observed.**
**2.2 Landslide distribution statistics and size-frequency analysis**
In sub-section 3.2, observed total landslide numbers and surface areas as well as other parameters
characterizing the statistics of the two inventories, the new one presented here for 2021 and the one for
2010 by Harp et al. (2016), are compared with 'predicted' ones. The latter numbers are computed
according to prediction laws proposed by Havenith et al. (2016) and Malamud et al. (2004). To estimate
the total number ($N_{LT}$, see Eq. 1) of landslides triggered by a specific earthquake, Havenith et al. (2016)
recommend to take into consideration the shaking intensity factor, (I, based on the Arias Intensity and
thus on the earthquake magnitude, M; see Eq. 6b in the next sub-section), the fault factor F (depending
on the type, FT, and size of the fault rupture, considering also the influence of a possible surface rupture),
the topographic energy (TE, using mainly as parameter the maximum altitude difference in the affected


region), the climatic background (CB) conditions, and the lithological factor (LF, depending on the
presence of soft soils for instance).
$N_{LT} = 1000 \times I \times F \times TE \times CB \times LF$ ,                    (1)
Compared with the prediction of the total number of landslides triggered by a specific earthquake
proposed by Havenith et al. (2016), the one recommended by Malamud et al. (2004) is much simpler (Eq.
2) and only based on the earthquake magnitude, M.
$N_{LT} = 10^{(1.29M - 5.65)}$ ,                    (2)
For the calculation of the total area potentially affected by landslides ($A_{Lext}$, area within the maximum
extent of landslide occurrence) Havenith et al. (2016) propose the following Eq. (3), which also directly
considers the earthquake magnitude, M, and the hypocentral Depth, D:
$A_{Lext} = I \times FT \times TE \times CB \times LF \times M \times D^2$ ,                    (3)
As Havenith et al. (2016), Keefer and Wilson (1989) also propose an equation to estimate the total area
potentially affected by landslides during one earthquake event. Their estimate of $A_{Lext}$ is purely based on
the earthquake magnitude, similar to Eq. (2) proposed by Malamud et al. (2004) to estimate $N_{LT}$ :
$A_{Lext} = 10^{(M - 3.46)}$ ,                    (4)
Malamud et al. (2004) do not propose any formula to estimate the total area potentially affected by
landslides during an earthquake event as Havenith et al. (2016) (see Eq. 3), but recommend the following
prediction law (Eq. 5) to estimate the total area effectively covered by co-seismic landslides, $A_{LT}$, based
on the observed or predicted (using Eq. 2, or any other related prediction law, such as the one in Eq. 1)
total number of landslides:
$A_{LT} = 0.00307 \, N_{LT}$ ,                    (5)
All the previous equations were used to compute the respective values presented in Table 1 in sub-section

255    3.2.

Size-frequency relations were computed for the 7091 landslide outlines in terms of frequency-density
function (FDF) on the basis of the measured surface areas, $f(A_L)$. The same statistics were also computed
for the 23,567 landslides mapped by Harp et al. (2016). Therefore, we used the method introduced by



Malamud et al. (2004) for surface areas (Eq. 6):
$$f(A_L) = \frac{\delta N_L}{\delta A_L} \qquad\qquad\qquad (6)$$
where $\delta N_L$ is the number of landslides with areas between $A_L$ and $A_L + \delta A_L$ (representing the difference
between two landslide surface area classes). Surface areas were calculated in $km^2$. Related distributions
computed, respectively, for each landslide catalogue (for the 2010 one by Harp et al., 2016; and for the
new 2021 inventory) are then compared with theoretical frequency-density distributions, as proposed by
Malamud et al. (2004). The latter are based on the three-parameter inverse-gamma probability
distribution (see equation 3 in Malamud et al., 2004) that is multiplied by the total number of landslides
of simulated events (100, 1000, etc.). In this regard, it should be noted that the original technique
proposed by Malamud et al. (2004) to complete the size-frequency statistics is based on the probability-
density values, corresponding to the frequency-density values divided by the total number of mapped
landslides, $N_{LT}$ (which can be fit by the aforementioned three-parameter inverse-gamma probability
distribution). However, as indicated above, due to the limited amount of high-resolution imagery
available for the area potentially affected by seismic shaking in August 2021, not all small landslides
could be mapped; therefore, the total number of landslides seismically triggered in August, $N_{LT}$, is likely
to be higher than 7091 (even if the potential 'hurricane-effect' is removed, as explained below), and the
probability-density function cannot be correctly computed. For such cases, Malamud et al. (2004)
recommend the computation of the frequency-density function to assess the completeness of the
inventory by comparison with the aforementioned predefined theoretical frequency-density functions, as
it will be shown for the 2010 and 2021 inventories in the following section.
**2.3 Mapping of seismic landslide triggering factors**
Above, we highlighted the climatic influence on slope stability in the target area that must be taken into
consideration when interpreting the landslide distribution statistics. However, it is obvious that for such
an event the main trigger factors are still related to earthquake shaking; those have to be assessed to
understand why extensive slope instability could be observed in one zone and only isolated minor failures
occurred in another one. Such an analysis is completed both for the 2010 and 2021 events, by computing
the Arias Intensity distribution maps (for 2010, comparing the results with the landslide distribution as



observed by Harp et al., 2016).
The Arias Intensity (Arias, 1970), Ia, can be considered as a quantitative measure of the degree of shaking
(in m/s) on the surface. With respect to any other intensity characterization (including the one based on
surveys) it has the advantage to be more objective and comparable for different earthquakes (according
to Harp and Wilson, 1995). Wilson and Keefer (1985) were the first to try to correlate seismically
triggered landslide distributions with this intensity measure. They also defined the following attenuation
relationship (Eq. 7a) in terms of magnitude (M) and hypocentral distance (R):
$$\log(Ia) = -4.1 + M - 2\log(R) + 0.5P \ , \tag{7a}$$
where P considers a possible deviation from the main law (P=0 stands for the average value).
Afterwards, Keefer and Wilson (1989) have reviewed the application of this formula and defined a new
one (Eq. 7b), for magnitudes greater than 7:
$$\log(Ia) = -2.35 + 0.75M - 2\log(R) \ , \tag{7b}$$
We applied the latter equation as both the 2010 and 2021 can be considered as M>=7 events. The R-
value represents the hypocentral distance map, here computed by using as source zone the blind fault
rupture segments of the 2010 and 2021 events (with 0 km epicentral distance and 10 km hypocentral
depth along the respective segment).
All equations introduced above have been applied to obtain the computation results presented below, in
the sub-sections 3.2 and 3.4.
**3    Results: landslide inventory statistics and analysis of trigger conditions**
This section first summarizes a series of landslide type and general distribution characteristics. Second,
landslide inventory and size-frequency statistics are presented and supported by an inventory
completeness analysis. Third, a study of possible climatic slope failure preconditioning and post-seismic
landslide surface changes is presented, which also compares landslide distributions with monthly
precipitation maps (using output maps of the Global Precipitation Measurement Mission, GPM, produced
by the NASA, for different periods, according to Acker and Leptough, 2007). Fourth, the landslide
occurrence observed in 2010 and in 2021 is compared with respective shaking intensity prediction maps.


### 3.1 Landslide type and distribution characteristics

Before analyzing specific statistical values of the two landslide inventories, the one compiled by Harp et al. (2016) for the 2010 event and ours completed after the August 2021 earthquake, we first have a look at the general respective spatial landslide distributions and provide basic information on the type of the mapped landslides.

The map presented in Fig. 5a shows that the global extent of landslides triggered in 2010 (brown outlines within the brown maximum extent polygon) and in 2021 (dark red outlines within the dark red maximum extent polygon) is quite similar (exact values are presented in Table 1). However, the 2010 landslide distribution appears a bit more dispersed than the one of 2021 that is marked by two focal areas, one in the southeastern part and one in the central western part of the total area affected in 2021. A major difference between the two landslide distributions can be observed with respect to the location of the EPGF zone. While most landslides occurred in the South of the fault zone in 2010, a relatively symmetric distribution of landslides with respect to the location of the EPGF zone can be observed for the 2021 event. This is mainly due to the fact that the fault segment that ruptured near EPGF in 2010 is located close to the coast (actually just in the South of the coast, as can be seen in the map in Fig. 5a), and thus only limited onshore surface areas could be affected by landslides in the North of the EPGF zone in 2010, while the location of the fault segment that ruptured in 2021 is more central within the southwestern peninsula of Haiti.

Another important observation is that there seems to be a gap between the zone affected by landslides in 2010 and the one affected in 2021. This means that, according to our present observations, the 2021 earthquake did not reactivate landslides triggered in 2010 – due to the large distance (> 60 km) between the fault ruptures. Actually, this point still has to be confirmed as the westernmost part of the area affected by the 2010 earthquake shaking is not covered (since August 14, 2021, until January 2022) by higher resolution imagery in GEPro. This check could only be done so far with the 10-m resolution Sentinel-2 imagery.
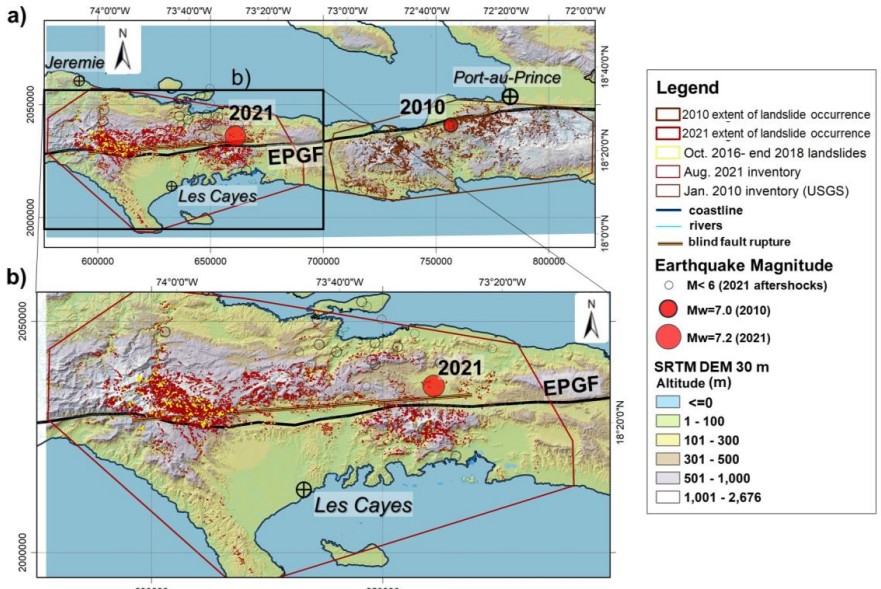

**Figure 5: a) Study region with areas affected, respectively, by the two Mw >= 7 events. Individual landslides triggered in 2010 (Harp et al., 2016, inventory) are mapped as small brown polygons (appearing as points at this scale) and the maximum extent of landslides triggered in 2010 is outlined by the large brown polygon. Landslides mapped for the 2021 event are shown as small dark red polygons within the maximum landslide extent area outlined by the large dark red polygon. b) Focus on the region hit by the August 2021 earthquake, with 7091 landslide locations shown by dark red polygons. See also main shock and after-shock (empty circles) location (from earthquake.usgs.gov) and outline of the (roughly 80 km long) blind fault rupture (extracted from USGS page: https://earthquake.usgs.gov/earthquakes/eventpage/us6000f65h/finite-fault).**

An important consequence of the specific location of the ruptured fault segments is that quite large landslides had occurred along the shore in 2010, where some of them had massively impacted the ocean and, thus, had produced up to 3 m-high Tsunami waves (see Olson et al., 2011; Poupardin et al., 2020; Fritz et al., 2013; Sassa and Takagawa, 2018) while there is not a single report of a major coastal landslide for the 2021 event – as the fault rupture occurred at a distance of minimum 10 km away from the nearest shoreline. Instead, a wider onshore area was exposed to high intensity earthquake shaking during the 2021 event. The related impact will be analyzed below on the basis of the statistical values presented in





Table 1.
Concerning the types of landslides triggered by the 2021 earthquake, we can say that by far most of them
can be classified as debris slides or flows (see examples in the GEPro view presented in Fig. 6b) and as
medium-size (most with a volume of less than 20,000 m$^3$) rockfalls (as shown above in Fig. 2). Thus, at
least 95% of all landslides mapped are relatively shallow (with a depth of less than 15 m). Actually, not
a single large massive landslide (> $2*10^7$ m$^3$) could be identified. A similar observation was made by
Harp et al. (2016) for the landslides triggered in 2010 (see view in Fig. 6c). However, when comparing
individual landslides induced in 2021 with those triggered in 2010, the latter are almost systematically
narrower than those of 2021 (compare the very narrow slides and flows in Fig. 6c with the typically
wider ones in Fig. 6b), while located in similar geological (limestone) and topographic (hilly-
mountainous) environments. Actually, in the so-called Ravine du Sud (Gorge of the South), part of which
is shown above in Figs. 3 and 4, numerous very extensive slope failures (but still relatively shallow)
could be observed; most of them formed by coalescent neighboring debris slides. Thus, entire slope units
(delimited by upper and lateral slope crests and the valley bottom) finally collapsed as one single mass
movement. Such kind of extensive slope failures occurred far less frequently in 2010 – at least onshore,
while at least a few aforementioned coastal and mostly submarine landslides must have been quite
massive as their impact had triggered Tsunami waves, as indicated above.
The fact that no really massive landslides had occurred (onshore), both in 2010 and 2021, also explains
why only a few longer lasting landslide dams had formed on the rivers. We could identify only about 100
minor dams (with a volume of less than 50,000 m$^3$, according to our estimate) after the August 2021
main shock, most of which had disappeared before the end of October 2021; and, only a few dozens of
them were impounding temporary lakes. In this regard it should be noted that Martinez et al. (2021), who
had also mapped landslides triggered by the 2021 Nippes earthquake (4893, according to their open file
report), have outlined almost 300 (at least partial) landslide dams after the event. However, they also
indicate that most of them failed a few days after formation; still, at the time of publication of their open
file report in December 2021, they consider 35 of the remaining dams as potentially hazardous. Here, we
will not further analyze this aspect as any related hazard assessment would require a site-specific
approach that is not targeted by this first study completed at regional scale.

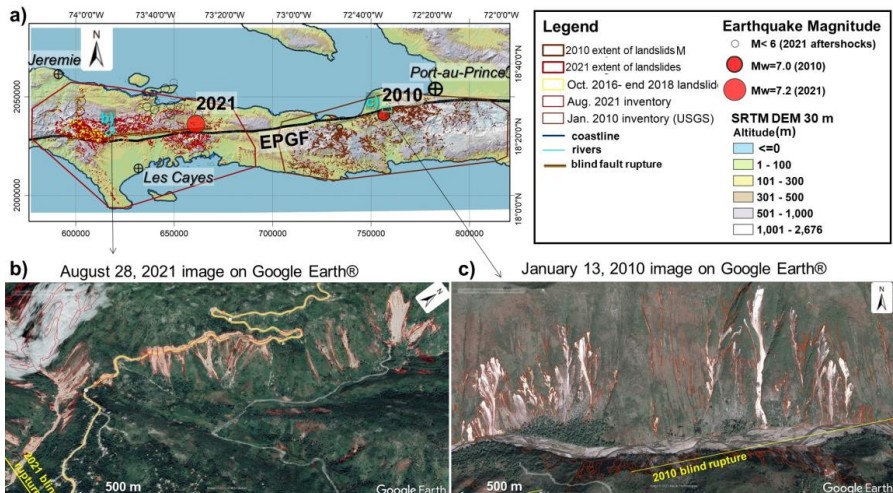


**Figure 6: a) Landslide distribution map for the two events in 2010 and 2021. b) GEPro view (© Google Earth Pro) of landslides triggered by the August 2021 earthquake. c) GEPro view (© Google Earth Pro) of landslides induced by the January 2010 main shock (with landslide outlines by Harp et al., 2016).**


While debris slides are the predominant type of 2021 slope failures in the central mountain ranges, widespread soft soil slides (but of smaller volume, typically of less than 10,000 m³) had occurred along the hills (with a crest altitude of less than 400 m) of the peninsula located in the Southwest of Les Cayes (southern part of map in Fig. 5b). As the slopes are very gentle, often seem to be less than 5°, it could be that those failures, many of which affected agricultural areas (marked by brownish disrupted fields), are related to liquefaction phenomena. However, also this observation has to be reexamined, by ground-control and site-specific studies, as the remote analysis based on 1-m resolution imagery does not allow us to fully confirm this interpretation.

**3.2 Landslide inventory and size-frequency statistics**

Table 1 presents an overview of landslide inventory statistics, for both the 2010 and 2021 events. The numbers in the first row show that apparently fewer landslides have been triggered in August 2021 (considering also the number of 4893 landslides published in the open file report by Martinez et al., 2021)



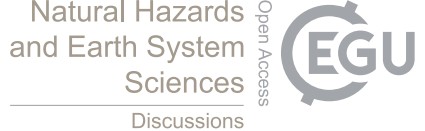

than in January 2010. At least two inventories, the one by Harp et al. (2016) and the one of Xu et al.
(2014), include far more landslide outlines (23,567 for the first, 30,828 for the second) than our catalogue
for 2021 (7091). Only the inventory by Gorum et al. (2013) that was the first one to be completed for the
2010 event contains fewer data (4490 points – not polygons - marking the landslide location). However,
paradoxically, a much wider surface area is covered by the apparently fewer 2021 landslides (a total area
of 84 km$^2$, see row 3 in Table 1) than by the more numerous 2010 landslides (sum of surface areas of
about 25 km$^2$, calculated for the Harp et al., 2016, inventory). This discrepancy can first be explained by
the fact that 2021 landslides could only be mapped from higher resolution imagery for about half of the
potentially affected area (in the eastern part). For the western zone, only Sentinel-2 images were available
until the end of 2021. Those 10-m resolution images typically do not allow for the (complete) mapping
of landslides smaller than 1000 – 2000 m$^2$. The second reason is documented by the GEPro views in Figs.
3, 4 and 6, showing that in 2021 many primarily individual landslides merged to form one single wider
mass movement – thus reducing the final number of single landslides; to this the aforementioned
'enlargement' effect of Hurricane Grace should be added, which might have contributed to the
coalescence of numerous 'small' landslides to form fewer larger landslides. Such a coalescence of
landslides seems to have occurred less frequently in 2010 (see parallel very narrow debris slides and
flows in Fig. 6c). In addition, for the coalescent debris slides, the type of mapping applied by Harp et al.
(2016) and most likely also by Xu et al. (2014) was different: they have created outlines for each single
component of a debris slide, even if those components form together a larger mass movement. To better
highlight the impact of the mapping technique and the availability of high-resolution imagery on the
landslide inventory completeness, a size-frequency analysis is presented in the second part of this sub-
section. Before, we first compare the observed landslide numbers with those predicted by Havenith et al.
(2016) and Malamud et al. (2014), respectively, for the two earthquakes. As introduced above (see Eq.
1), according to Havenith et al. (2016), this number depends on the seismic intensity (I, using as input
the Ia value computed for the respective earthquake magnitude), the fault factor (type, size and possible
surface rupture), the topographic energy (maximum difference of altitudes in the affected area), the
climatic background (in this case marked by tropical wet climate), and the lithological factor (here using
an average type, for rocks in general). For the precise classification of the different factors, the reader is
referred to Table 1 in Havenith et al. (2016). Here, we used the values presented below in Table 2





(considering both events in 2010 and 2021), which indicate the following:
1) the shaking intensity values, I=0.74, in 2010, and I=1 in 2021 are characteristic for the respective
magnitudes (note, this factor can reach a value of up to 3.5 in the case of high-magnitude earthquakes,
with Mw > 8);
2) the fault factor, F=2.25, can be considered as similar in both cases, marked by an oblique slip that
occurred along a fault segment with a length of 50-100 km, with no clear surface rupture (note, F can
reach a value of up to 6 in the case of a surface rupture of an activated reverse fault segment with a length
of more than 300 km, such as observed for the Wenchuan earthquake in 2008);
3) the topographic energy value, TE=2, in both cases characterizes a surface morphology marked by local
altitude changes of more than 500 m within a hilly region (only smaller mountains, with an altitude of
less than 2500 m can be found in the regions affected by the 2010 and 2021 events; note, Havenith et al.,
2016, selected a value 4 to mark the high steep slopes in the Longmenshan Mountains affected by the
Wenchuan earthquake in 2008);
4) the climatic background factor, CB=1.5 marks relatively wet conditions for the 2021 event while CB=1
indicates average conditions for the 2010 event (the higher value chosen for 2021 considers some
preconditioning of slope instability by Hurricane Matthew, as explained in the next section; note,
Havenith et al., 2016, selected a value CB=2 for the very wet conditions that can be found in the
Longmenshan Mountains affected by the Wenchuan earthquake, characterized by yearly precipitation
values of more than 3000 mm – while the target areas in Haiti are marked by values of about 2000 mm);
5) the lithological factor, LF=2, indicates that both weathered rocks and soft soils can be found in the
affected area (note, Havenith et al., 2016, selected a maximum value, LF=4, for the Haiyuan-Gansu-
Ningxia earthquake event, China, in 1920, as it affected an area that is almost entirely covered by Loess
deposits, which are highly susceptible to slope failure).
When these different factor values are combined according to Eq. (1) presented above, the total numbers
of landslides, $N_{LT}$, predicted for the 2010 and 2021 events are, respectively, 6694 and 13,476. These
values can be compared with the numbers predicted by the simple equation (Eq. 2), proposed by
Malamud et al. (2004), using only the earthquake magnitude as input: 2399 for the 2010 event and 4345
for the 2021 event. The latter prediction seems to clearly underestimate the observed numbers of
triggered landslides, while those predicted by using Eq. (1) by Havenith et al. (2016) provide intermediate





values: larger than the number observed by Gorum et al. (2013) but smaller than the numbers observed
by Harp et al. (2016) and by Xu et al. (2014). The two predictions (Eq. 1 and 2) were also applied to the
2021 event; the first one producing a higher $N_{LT}$ (=13,476) than the observed value, the second one
producing a lower value (=4345).
As shown on the maps in Fig. 5, also the total area within the maximum extent of landslide occurrence,
$A_{Lext}$, was outlined and then measured for the 2010 and 2021 events. The latter is about 1.4 times larger
than the first: 5100 km$^2$ for 2021 against 3700 km$^2$ for 2010. These values can be compared in Table 1
with the predictions by Havenith et al. (2016) and by Keefer and Wilson (1989), corresponding,
respectively, to 3124 and 3467 km$^2$, for the 2010 event, and to 6470 and 5495 km$^2$, for the 2021 event.
In this case, the very simple equation proposed Keefer and Wilson (1989) provides an estimate of $A_{Lext}$
that is closer to the observed value than the one produced by the more complex relationship proposed by
Havenith et al. (2016).
The third row of Table 1 compares the total observed slope areas affected by landslides, $A_{LT}$,
corresponding, respectively, to a value of 24.86 km$^2$ for the 2010 event and of 84.38 km$^2$ for the 2021
event, with the values predicted by Eq. (5) by Malamud et al. (2004) for each event. For 2010, we applied
this relationship to the three observed values indicated in the first row and by using the $N_{LT}$, predicted
respectively by Havenith et al. (2016) and Malamud et al. (2004). Among all total landslide surface area
values predicted for the 2010 event, it can be seen that the one based on the Havenith et al. (2016) $N_{LT}$
estimate produces the best fit (=20.55 km$^2$) when compared with the observed value of 24.86 km$^2$. For
2021, the respective predictions all underestimate the observed total landslide surface area value, $A_{LT}$, by
a factor of at least two, even when the highest $N_{LT}$ estimate (using Eq. 1) by Havenith et al. (2016) is
used.
The fourth and fifth rows show the average and median landslide surface area values, respectively, for
the 2010 landslide inventory by Harp et al. (2016) and the new 2021 inventory. Those values are clearly
higher for the last inventory, confirming on the one hand that larger landslides were triggered in 2021,
and, on the other hand, that many more small landslides were mapped by Harp et al. (2016) than by our
team for 2021. Especially the large difference between the median landslide surface area values (the 2021
value is almost twenty times larger than the 2010 value) highlights the 'mapping-related' factor, which
becomes obvious when considering the values in the next four rows. On the one hand, the smallest



landslide mapped by Harp et al. (2016) has a surface area of 0.5 m$^2$ and their inventory contains 6587
landslide polygons smaller than 100 m$^2$ while our inventory for 2021 only includes one landslide with a
surface area smaller than this value. On the other hand, the largest landslide mapped for the 2021 event
(>400,000 m$^2$) has almost twice the size of the largest one that occurred in 2010, when actually only 2
landslides larger than 100,000 m$^2$ had been triggered; in 2021, we could outline more than 100 landslides
larger than this value.
Finally, Table 1 provides information about the distribution of the 2010 and 2021 landslides with respect
to the blind fault rupture projected on the surface (near the EPGF outline). As already introduced above,
a much larger number of landslides occurred in the North of the latter in 2021 (=4678) compared to 2010
(=2548, at least for onshore slope failures); consequently, more landslides occurred in 2010 in the South
of the respective blind fault rupture. As the total number of mapped landslides is much larger for the
2010 event, the difference between those numbers is very high: 21,019 occurred in the South of the fault
rupture in 2010 and only 2420 in the South of the respective fault rupture in 2021. However, when the
total surface area affected by landslides is considered, the 2021 event affected more zones both in the
South and the North of the fault rupture than the 2010 event, while the distribution of landslides for each
event with respect to the fault rupture remains the same also when considering the affected surface areas:
they are much larger in the South of the fault rupture for the 2010 event but larger in the North for the
2021 event. The main explanation for this difference has already been provided above: the fault segment
that ruptured in 2010 is located close to the coast, with limited onshore surface areas being exposed to
landslide activity in the North of the respective fault rupture, while the location of the fault rupture in
2021 is more central with respect to the shorelines of the southwestern peninsula of Haiti. Actually, the
presence of more numerous and larger landslides in the North of the fault rupture of 2021 could be
expected, according to the observations made by Fan et al. (2018) for the Wenchuan earthquake in 2008,
which had triggered far more landslides on the hanging wall of the activated reverse fault segments than
on the foot wall. Considering the oblique slip character along the fault ruptures of 2010 and 2021 dipping
to the North, the hanging wall is located on the northside of the blind fault rupture projected on the
surface and would logically host more landslides (as indeed observed for the 2021 event). Thus, if a
larger 'onshore hanging wall' surface area (marked by a hilly or mountainous morphology) had been
available onshore for the 2010 event, it can be assumed that even more landslides would have been





triggered (onshore). Now, we can only assume that in addition to the few known coastal failures also
massive submarine landslides must have occurred in the North of the 2010 fault rupture.

**Table 1: 2010 and 2021 landslide inventory characteristics – where not specified for the 2010 event, using the**
**Harp et al. (2016) inventory. The largest values for each specific observation/estimate (if more than 1 indicated)**
**are bold.**

| Landslide inventory parameters/predictions | 2010, Mw=7.0 | 2021, Mw=7.2 |
|---|---|---|
| Observed number of landslides, $N_{LT}$ | >4490[a] / 23,567[b] / **30,828**[c] | 7091/4893[d] |
| Havenith et al. (2016) $N_{LT}$ prediction 1 | 6694 | 13,476 |
| Malamud et al. (2004) $N_{LT}$ prediction 2 | 2399 | 4345 |
| Area of region potentially affected by landslides, $A_{Lext}$ (km$^2$) | 3700 | 5100 |
| Havenith et al. (2016) $A_{Lext}$ prediction 1 | 3124 | 6470 |
| Keefer and Wilson (1989) $A_{Lext}$ prediction 2 | 3467 | 5495 |
| Total surface area of landslides, $A_{LT2}$ (km$^2$) | 24.86 | 84.38 |
| Malamud et al. (2004) $A_{LT}$ prediction : | | |
| for the observed $N_{LT}$ | 13.8[a] / 72.3[b] / **94.6**[c] | 21.8 |
| for the $N_{LT}$ prediction 1 | 20.55 | 41.4 |
| for the $N_{LT}$ prediction 2 | 7.36 | 13.3 |
| Average area (m$^2$) | 1055 | 11,886 |
| Median area (m$^2$) | 254 | 4729 |
| Smallest landslide (m$^2$) | 0.5 | 75 |
| Number of landslides smaller than 100 m$^2$ | 6587 | 1 |
| Largest landslide (m$^2$) | 234,370 | 409,479 |
| Number of landslides larger than 100,000 m$^2$ | 2 | 103 |
| Total number of landslides in the North (N) / South (S) of the fault rupture | N= 2548  **S= 21,019** | **N= 4678**  S= 2420 |
| Total surface area of landslides in the N / S of the fault rupture (km$^2$) | N= 2.45  **S= 22.41** | **N= 58.31**  S= 26.07 |

[a] Number of landslides observed by Gorum et al. (2013), [b] by Harp et al. (2016), [c] by Xu et al. (2014),
and [d] by Martinez et al. (2021).

In addition to the numbers shown in Table 1 and explained above, we also provide two values for the



smaller landslide inventory compiled for the period between October 10, 2016 and the end of 2017. For
this period, 324 landslide zones have been outlined (see yellow polygons shown on the views and maps
in Figs. 3 and 5), covering a total surface area of 7.92 km$^2$. However, we must indicate that these values
represent approximations as the main focus was on the identification and the determination of the general
extent of those pre-2021 slope failures rather than on their detailed mapping.
**Table 2: Factors contributing to the total number and surface area of landslides triggered by the 2010 and the**
**2021 earthquakes, according to the prediction proposed by Havenith et al. (2016). The minimum and**
**maximum values proposed by Havenith et al. (2016) are also indicated, the latter with information on the**
**event – region, to which this maximum factor value was attributed.**

| Haiti Events/ Factors | Shaking Intensity, I | Fault Factor, F (type, FT) | Topographic Energy, TE | Climatic Background, CB | Lithological Factor, LF | Hypocentral Depth, D (km) |
|---|---|---|---|---|---|---|
| 2010 | 0.74 | 2.25(1.5) | 2 | 1.5 | 2 | 10 |
| 2021 | 1 | 2.25(1.5) | 2 | 1 | 2 | 10 |
| min. values | 0.1 | 0.75 | 1 | 0.5 | 1 | 10 |
| max. values (event - region) | 3.4 (Chile, 1960) | 6 (Wenchuan, 2008) | 4 (Wenchuan, 2008) | 2 (Wenchuan, 2008) | 4 (Haiyuan-Gansu-Ningxia, 1920) | 226 (Hindu Kush, 2002) |


Considering the values presented in Table 1, we still have to explain why the total surface area covered
by landslides in 2021 is much larger than the one covered by the 2010 landslides: a) the first obvious
physical reason for the larger area hit by mass movements in 2021 is the higher triggering earthquake
magnitude of the last event (this effect will also be analyzed below by comparing the influence of shaking
intensity on landslide susceptibility); b) another physical reason could be the possibly higher
susceptibility to mass movements of the western part of the peninsula hit by the 2021 event, compared
to the eastern part (this factor still has to be analyzed on the basis of landslide susceptibility maps,
considering also the geological influence, which have been computed and will be presented in another
paper); c) a third reason for the larger area affected by landslides in 2021 could be related to the
aforementioned 'hurricane' effects that will be analyzed in the following sub-section; d) and fourth, the
more central location of the fault segment activated in 2021 with respect to the coasts of the peninsula
could also explain the larger (subaerial) slope failures triggered during the last event within the wider

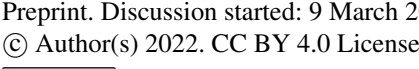


onshore hanging wall part, as already introduced above.
However, we also have to admit that a small percentage of the larger landslide surface area mapped for
the 2021 event could be related to the aforementioned technique of outlining coalescent landslides as
single ones, where small unaffected areas have been included within the landslide polygon. For instance,
this 'over-mapping' of landslide areas became obvious after the first rapid mapping session, only based
on Sentinel-2 imagery. Thus, during the later refinement and splitting of the landslide polygons (using
the higher resolution imagery that became available on GEPro), wide surface areas initially mapped
within the landslide polygons were then excluded from them. Interestingly, the total surface area could
not be reduced by this refinement (the initial total surface area covered by landslide polygons was about
75 km$^2$, which increased to 84 km$^2$, as shown in Table 1), as during the second mapping phase also
numerous new smaller landslides could be mapped, the total area of which more than compensated the
reduced area of the refined initial landslide polygons.

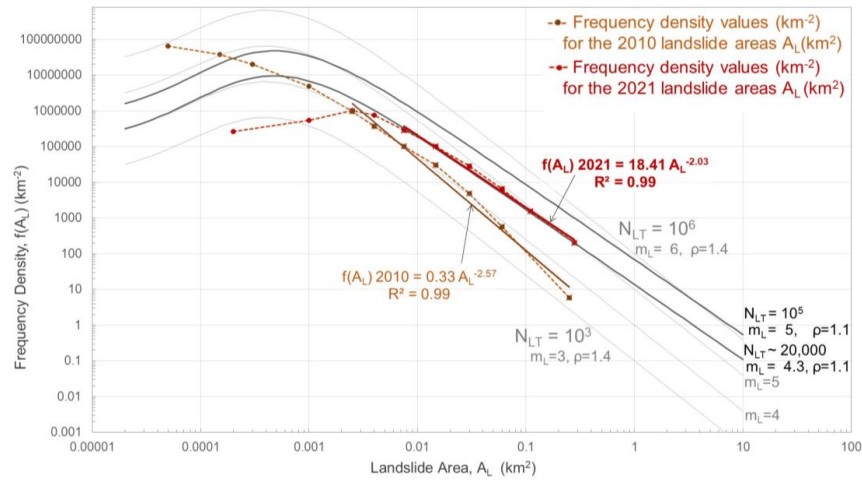


**Figure 7: Frequency density graphs developed for the 2010 (in brown, by Harp et al., 2016) and the new 2021**
**landslide inventories (in dark red), compared with landslide frequency-density curves computed for different**
**landslide event magnitudes according to equation 3 in Malamud et al. (2004). For computing the light gray**
**curves, the decay exponent determined by Malamud et al. (2004) was used (=-2.4, = -(ρ+1)), while for the dark**
**gray curves an exponent of -2.1 was used (similar to the one obtained for the 2021 landslide frequency-density**
**distribution).**




To better quantify the 'mapping effect' on the landslide distribution statistics, we carried out an inventory
completeness analysis as proposed by Malamud et al. (2004), by computing and plotting frequency-
density values for various landslide surface area classes as shown on the graph in Fig. 7. Related curves
are compared with theoretical frequency-density distributions computed for various simulated landslide
events, composed, respectively of 1000, 10,000, 100,000 and 1,000,000 elements, here marked by the
landslide event magnitudes, $m_L$ (3 to 6), corresponding to the logarithm (with base 10) of these values.
There are two important parameters to be analyzed for the observed frequency-density distributions,
through comparison with the theoretical curves: the first part is represented by the power-law decay (see
introduction in Stark and Hovius, 2001) that appears as a linear decay in the log-log graph below; the
second part is the so-called 'rollover', which can be observed for a landside surface area where the
exponentially decreasing number of larger landslides turns into a decrease of the number of smaller
landslides. According to Malamud et al. (2004) the main parameters characterizing these two parts should
be identical for all landslide events: the decay of larger landslide numbers should have an exponent-value
of about -2.4 (=-(ρ+1) for ρ=1.4, being the parameter controlling the power-law decay), which was used
for the calculation of the light gray theoretical frequency-density curves, while the rollover should occur
for all landslide events within the same landslide size class, marked by a surface area of about 400 m².
While a power-law decay can indeed be observed for both landslide inventories, the exponents
characterizing this decay slightly differ from the value of -2.4 that Malamud et al. (2004) had determined
for other inventories: for the 2010 inventory, the related absolute value is slightly higher (-2.57, for the
brown line fitting the 2010 data) and for the 2021 inventory it is clearly lower (-2.03, for the red line
fitting the 2021 data). Concerning the rollover, the 2010 landslide frequency-density distribution
interestingly does not present any such feature, while the 2021 inventory is marked by a rollover that
occurs for a landslide size class of about 3000 m². The comparison of this value with the much smaller
value determined by Malamud et al. (2004) clearly hints at the incompleteness of the 2021 inventory, at
least for the smaller landslides. To estimate this incompleteness, the part of the power-law decay
(supposed representing the 'complete' part of the landslide inventory) has to be compared with the
theoretical frequency-density curves of the simulated landslide events with various magnitudes (light
grey lines in the graph in Fig. 7). According to this comparison, the 2021 landslide event would have a

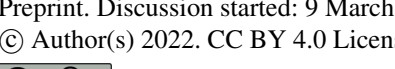



magnitude, $m_L$, close to 5 (=100,000 landslides!). For the 2010 event, the same comparison indicates a
magnitude, $m_L$, close to 4 (=10,000 landslides!). While the theoretical landslide event magnitude of the
2021 event seems to largely overestimate the observed number of landslides, the one obtained for the
2010 event seems to underestimate the observed total number of landslides. Actually, the multiple
predictions presented in Table 1 for the 2010 event also underestimate the high numbers of landslides
observed by the two teams of Harp et al. (2016) and of Xu et al. (2014). Combining this comparison with
the fact that the landslide inventory of Harp et al. (2016) does not present any rollover, and includes even
more than 6000 landslides with a surface area smaller than 100 $m^2$, this catalogue (and possibly also the
one of Xu et al., 2014) can be considered as 'superfine' – at least, when compared with the type of
landslide distributions (and related mapping of individual slope failures) analyzed by Malamud et al.
(2004). And this 'subtle refining' of the landslide outlines, partly related to the separate mapping of
landslide zones that are coalescent, may also explain the higher exponent value of the power-law decay
of the 2010 landslide frequency-density distribution as well as the high number of mapped landslides
compared to all predicted values (see Table 1). In contrast, the rollover observed for a much larger
landslide size class than the one predicted by Malamud et al. (2004), the smaller decay exponent (-2.0
instead of -2.4) obtained for our 2021, and the smaller observed total landslide number values compared
to the one predicted by Havenith et al. (2016), all confirm the incompleteness of the latter catalogue, at
least for landslides smaller than 3000 $m^2$. In the discussion part, some tools to remediate this problem in
the future will be presented. However, as proposed by Malamud et al. (2004), by comparing the
frequency-density distribution with the theoretical curves, an estimate of the actual total number of
landslides could be made. As introduced above, the one of the 2021 event would be close to 100,000.
Here, we will first propose a comparison with the same type of theoretical frequency-density curves, but
computed for a decay exponent of -2.1, similar to the one of decay of the observed 2021 landslide
frequency-density distribution. The graph in Fig. 7 shows that the 2021 data are best fit by such a curve
representing an inventory of 20,000 landslides. Actually, a similar value could also be obtained for an
exponent of -2.4 if we consider a lower number of large landslides that would in fact be composed of
smaller ones (the weight of the latter would then increase and the one of the larger ones decrease, which
leads to a stronger decay). Below we will discuss, which assumption would be the best: (1) that really
100,000 landslides might have occurred in 2021, or (2) that the decay exponent value of -2.4 is indeed



not as 'universal' as proposed by Malamud et al. (2004), or (3) that a future refinement of the 2021
landslide inventory would result in a frequency-density distribution marked by a higher (absolute value
of the) decay exponent, close to -2.4.
**3.3 Climatic (pre- and post-seismic) conditioning of slope instability**
The climatic influence on landslide occurrence (in 2021) has been introduced above, by considering the
possible impacts of hurricanes on slope failure occurrence, marked both by preconditioning of slope
instability and by post-seismic intensification. We first start analyzing the last effect, by considering the
potential impact of Hurricane Grace on post-seismic landslide intensification, on August 16-17, 2021
(when it had crossed the target region and was actually classified as tropical depression at that stage). A
possible effect of related rainfalls on landslide occurrence has already been highlighted, for instance, on
the AGU Landslide blog (by Petley, D., 2021, on blogs.agu.org/landslideblog). This effect could be
confirmed when we compared Sentinel-2 imagery collected right after the earthquake (2h after the main
shock) with images remotely sensed after August 17, 2021. As indicated above and shown in Fig. 4, an
intensification of denudation could indeed be observed after the tropical storm Grace event. However,
one important limitation has to be highlighted: this comparison could only be completed for about 10%
of the area potentially hit both by the earthquake and by Grace, due to the intense cloud cover present in
the target region during that period. Furthermore, another effect could have contributed to slope failure
intensification after the main shock on August 14, the one related to the aftershocks (see empty circles
shown in all maps above), but analyzing this effect would require a refinement of the satellite image
analysis both in space and time, which is hardly possible considering the extensive cloud cover present
in the target area when all those seismic shocks occurred. Here, we will focus on the possible climatic
influence, which can better be outlined when comparing the landslide distribution with actual
precipitation maps. Therefore, we used the Global Precipitation Measurement Mission (GPM, by NASA)
data obtained via the https://giovanni.gsfc.nasa.gov/ website, corresponding to the merged satellite-gauge
monthly precipitation estimate (in mm), assessed with a resolution of 0.1°. Related maps were requested
for all months between August 2000 and July 2021, and also for the specific months of October 2016
and August 2021, as well as for all October months between 2000 and 2020. However, we need to
indicate that these rainfall estimates could, unfortunately, not be confirmed by ground measurements due



to missing data availability (information received by the Centre National de l'Information Géo-Spatiale,
CNIGS, of Haiti). Fig. 8 presents the three first types of maps, while the last one is compared with the
first and third type in Fig. A1, in the annex. By comparing the merged satellite-gauge precipitation
estimate for August 2021 (Fig. 8b) with the monthly precipitation map averaged for all months of the
previous 20 years (Fig. 8a), we can clearly see that August 2021 was indeed marked by a higher
precipitation rate, which is most likely related to the Grace event. However, the most intense precipitation
did not affect the region hit by the 2021 earthquake but the eastern part of the peninsula, roughly covering
the same region as the one affected by the 2010 event (note, we did not check any landslide reactivation
after Grace for that area). The region hit by the 2021 earthquake was not affected by much higher monthly
precipitation rates than usual: for the central seismically affected zone, in the North of Les Cayes, about
240-280 mm had been recorded in August 2021, against a monthly average of 200 mm. Thus, just by
considering these data, one would not expect an important climatic contribution to slope failure
occurrence in the region affected by the 2021 earthquake. Still, an influence could be observed and this
is likely to be related to the concentration of most of the 'monthly precipitation' of August 2021 within
the two days (Aug. 16 and 17) of the Grace tropical storm event, just two days after the 2021 main shock.
As indicated above, we estimate that related precipitation has resulted in an increase of landslide surface
areas of about 10-15%. Due to the limited extent of zones where this check can be made (only considering
the cloud-free areas on the Sentinel-2 image of August 14, 2021), it was decided to map all areas covered
by landslides after August 14, 2021, also those which are likely to have been (re)activated by rainfall –
the total effect of which can barely be controlled and quantified outside the 10% of cloud-free zones
visible on the image collected right after the main shock. The only 'correction' that can be made is to
reduce the total surface area mapped as landslides by those 10-15% to estimate the one that was actually
affected by co-seismic slope failures: thus, instead of considering the value of 84 km$^2$, it is possible that
co-seismic landslides covered a total surface area of 'only' 75-78 km$^2$ – which is still three times more
than the total surface area covered by 2010 co-seismic landslides (close to 25 km$^2$).
To explain this great difference between the total surface areas, we will analyze the possibility of a
preconditioning of slope instability due to climatic events that occurred before August 2021. Therefore,
we compare the average monthly precipitation rates between 2000 and 2021 (Fig. 8a) with the one of
October 2016 (Fig. 8c). For that month, a peak of intensity of 626 mm can be observed for the area


between Gran Rivière De Nappe and Petite-Rivière-de-Nippes, situated immediately in the North of the
epicentral area of the 2021 main shock. Actually, the whole area potentially affected by the August 2021
earthquake had been exposed to abnormal precipitation rates of more than 400 mm in October 2016 (to
be considered as abnormal also when comparing with the average precipitation of all months of October
between 2000 and 2020, of 200-320 mm, as shown in Fig. A1). For October 2016, those values were also
the highest ones compared with the rest of the country; this clearly indicates that they must be related to
a specific climatic event, which can easily be identified as Hurricane Matthew that had crossed the
western peninsula (including the region hit later by the August 2021 earthquake) on October 4-5, 2016.
And, precisely for this region that had been exposed to abnormal precipitation rates in October 2016, we
could outline 324 pre-seismic landslides (yellow polygons shown above in the maps in Figs 1, 3 and 5
and below in Fig. 8), mapped for the period between mid-October 2016 and the end of 2017. In this
regard, it should be noted that we had to extend the observation period (beyond the end of 2016), as not
all regions were cloud-free right after the hurricane event or are still not covered by higher, 0.5-1 m,
resolution imagery available for that period. Outside this region, no (or only very few) clear landslide
activations could be identified between mid-October 2016 and the end of 2017. And, practically all 324
landslide zones (at least 90% of them) are located within those mapped for the August 2021 seismic event
(which are still marked by a much higher level of denudation compared to the October 2016 activation).
These observations allow us to conclude that the Hurricane Matthew event has preconditioned slope
instability over the region hit later by the August 2021 earthquake. This preconditioning factor could also
explain why three times larger surface areas have been affected by landslides in 2021 compared with
2010. Further, it must be added that the 2010 earthquake had not been preceded by any particular
hurricane event during the previous ten years, at least not by any storm that had caused abnormal
precipitation amounts specifically within the region hit by the 2010 earthquake. However, in the
discussion, we will also consider a general influence of tropical storms on the distribution of the
landslides triggered in 2010 (and also for those triggered in 2021, in addition to the Hurricane Matthew
effect), notably to explain why numerous landslides had occurred very far from the seismic source zone.


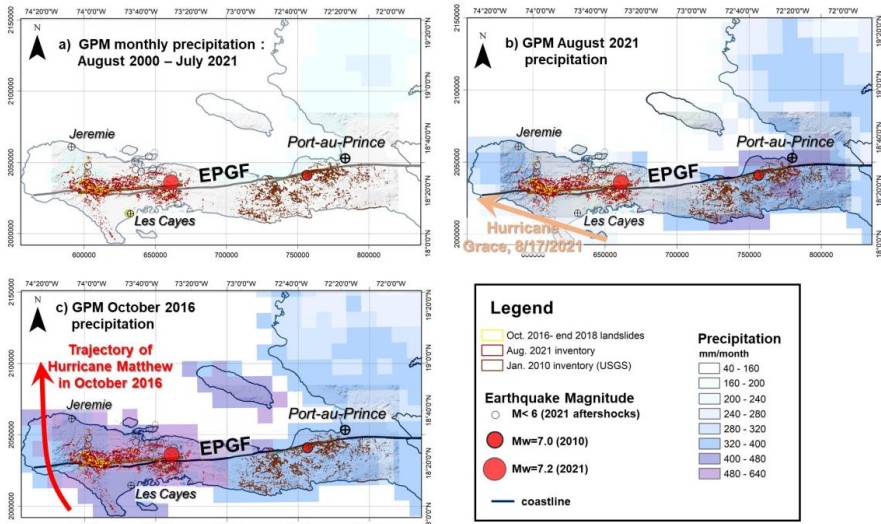


**Figure 8: Monthly Global Precipitation Measurement Mission (NASA) maps (0.1° resolution,**

**values in mm/month) for southwestern Haiti, (a) for all months between August 2000 and July 2021,**

**(b) for August 2021 (with indication of the track of Hurricane Grace – at that stage still to be**

**considered as a tropical storm), and (c) for October 2016 (marked by the Hurricane Matthew**

**event).**

**3.4 Shaking intensity maps**

Above, we fist analyzed the possible climatic influence on seismically induced slope failures as it could

affect the landslide distribution and thus has to be taken into consideration when assessing and

interpreting the seismic effect on landslide occurrence. The latter will only be analyzed here at regional

scale. Therefore, we compare the landslide distributions observed for the 2010 and 2021 events with the

respective estimated Arias Intensity (Ia) attenuation maps, computed by applying Eq. (7b) introduced

above, as recommended by Wilson and Keefer (1985) and also by later studies (e.g., Harp and Wilson,

1995, among many others). The map in Fig. 9a presents the 2010 and 2021 mainshock Ia attenuation

values, with a maximum shaking intensity of 11.2 m/s computed for the 2021 event and 7.9 m/s for 2010

(respective maps are partly overlapping in the central region, but not summed up, keeping the individual

values). This map shows that all 2010 and 2021 landslides are included within a zone marked by an Ia


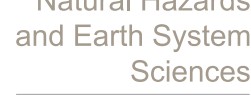
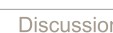

threshold of 0.2 m/s (close to the one proposed by Keefer and Wilson, 1989, for disrupted slides and
falls). Actually, for 2021, 99% of the total landslide surface areas are even located within a zone marked
by Ia values lager than 1 m/s; however, only 80% of the total surface areas of the 2010 landslides are
included within the respective Ia >= 1 m/s zone. Thus, the latter mass movements appear as more
'dispersed' with respect to the estimated seismic intensity attenuation than the 2021 ones.
Notwithstanding this dispersion, and the overlap of Ia values larger than 0.2 m/s in the central zone
between the two blind fault ruptures of 2010 and 2021, not a single landslide of 2010 seems to have been
reactivated in 2021. This observation raises the question if the central 'landslide gap' is due to an
overestimation of the Ia values in this central zone (as this zone is marked by Ia values above the
aforementioned minimum threshold of 0.2 m/s, for both events, and thus should have been affected by
landslides both in 2010 and 2021, according to the shaking intensity prediction parameter), or if this zone
is simply less susceptible to (seismic) slope failures.

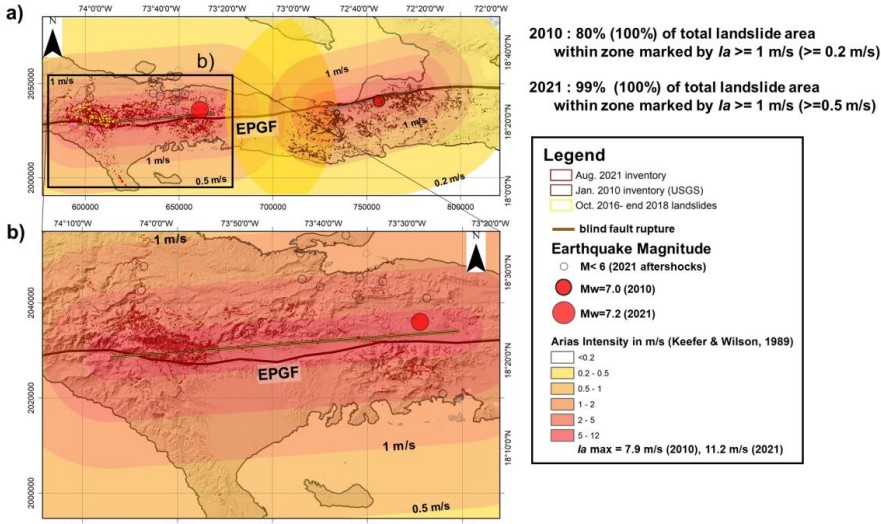

**Figure 9: a) Arias Intensity (Ia) attenuation maps computed (by using Eq. 7b, by Keefer and Wilson, 1989)**
**for the 2010 and 2021 main shocks in Haiti; see also indication of % of total surface area of landslides observed**
**for different Ia thresholds. b) Focus on the respective map computed for the 2021 event.**

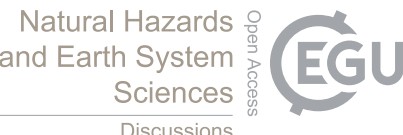
To answer this and other related questions, a full landslide susceptibility analysis has been completed
and will be presented in another paper. Here, only the possible links between landslide distribution the
aforementioned seismotectonic and climatic factors will be discussed.
**4    Discussion**
From the comparison of the two landslide catalogues (2010 and 2021), we could infer that apparently
not a single landslide triggered in August 2021 occurred within the zone previously impacted by the 2010
event. There is a gap of about 10 km between the westernmost 2010 and the easternmost 2021 landslide
(see gap between the general outlines of the maximum extent of landslides triggered in 2010 and in 2021
shown on the map in Fig. 5a). Thus, we assume that there was no obvious preconditioning of landslide
generation in 2021 by the 2010 event, while landslide studies completed in other parts of the World (e.g.,
by Parker et al., 2015, for events in New Zealand) could outline an influence of previous earthquakes on
landslide occurrence during later events. The absence of this influence by the 2010 earthquake is
probably due to the long distance (the 'gap') of about 60 km between the fault segments that ruptured in
2010 and in 2021. However, by citing Saint Fleur et al. (2020), Stein et al. (2021) hint at an older event,
of 1770, with an assumed magnitude of 7.5 and an epicenter located precisely in the gap between the
2010 and 2021 blind fault ruptures, which could also have affected the region hit by the 2021 earthquake.
At present, we cannot exclude that this older event had preconditioned some slopes (by soil weakening,
rock fracturing) affected by some larger landslides in 2021; however, very shallow slope failures initiated
in 1770 are unlikely to have stayed in place over such a long period of more than 250 years, as they
would have been 'washed' away by the next tropical rains.
Second, none of the two earthquakes triggered very massive landslides, such as deep-seated rockslides
with a volume of more than 20 $10^6$ m$^3$ (while extensive areas are covered by layered and weathered
limestone that could also produce massive slope failures; the related geological influence on landslide
occurrence will be analyzed in the landslide susceptibility paper presently under preparation). Such
massive failures have been observed after many M7+ events in other mountainous regions of the world:
see Fan et al. (2018) for the 2008 Mw=7.9 earthquake in China, or Havenith et al. (2015) for a series of
M>7 events that hit Central Asian mountain regions during the last 120 years. This is partly due to the

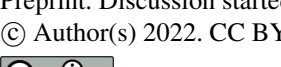



fact that the regions hit by the two earthquakes in Haiti are covered by mountains of limited elevation
changes, typically less than 1000 m – while, for instance, the Longmenshan Mountains hit by the 2008
Wenchuan earthquake, present elevation changes of up to 3000 m over relatively short (<6 km) distances
(Fan et al., 2018). This fact, combined with the higher magnitude of the Wenchuan earthquake (Mw=7.9),
could partly explain the much larger number of massive rockslides triggered by the latter event in China.
As third general observation, we highlight the fact that the 2010 event triggered most landslides in the
South of the activated fault segment, while in August 2021 about 2/3 of all landslides were triggered in
the North of it, within the hanging wall (according to the fault mechanism provided by the USGS
Earthquake Hazard Program page, earthquake.usgs.gov). In this regard, the Wenchuan earthquake has
clearly marked the effect of the hanging wall on the landslide distribution: about 90 % of all landslides
were triggered on top of the reverse fault dipping towards the West-Northwest, only a minor portion
occurred on the more 'stable' foot wall (Gorum et al., 2011; Fan et al., 2018). The 'hanging wall effect'
on landslide triggering can be explained by stronger upward oriented shaking that contributes to a higher
surface acceleration and more intense slope failures; additionally, all (or most of the) aftershocks
occurred within the hanging wall, increasing the seismic shaking intensity cumulated over the active
seismic period in the related surface area, which could have contributed to prolonged landslide activity
as well (to be added to the climatic effect introduced above and discussed below). This effect may thus
also be at the origin of the more widespread landslide occurrence in the North of the 2021 blind fault
rupture. The reduced number of landslides induced on the hanging wall side of the 2010 fault rupture can
be explained by the absence of high and steep slopes (onshore) on this side. Actually, a few known
massive landslides occurred near the coast, but are mostly located on submarine slopes in the 2010
hanging wall zone. Three of them reportedly also caused Tsunami waves (see Olson et al., 2011, among
others) – a phenomenon that was not observed for the 2021 event, as the coasts are located farther away
from the seismic source zone.
In section 2, we have outlined the two different types of landslide mapping techniques applied to the
2010 event (by Harp et al., 2016) and to the August 14, 2021, earthquake (the new inventory presented
here). Our somewhat 'rougher' mapping technique adapted to the lower resolution imagery immediately
available right after the 2021 earthquake first explains the much smaller number of landslides (7091)
mapped around the fault segment that ruptured in 2021, compared to the 2010 event (>23,000 landslides



mapped by Harp et al., 2016). Second, we acknowledge that the mapping of coherent landslide zones
compared with the outlining of individual landslide sources and flows by Harp et al. (2016) can result in
'over-mapping' of large landslides and, thus, in decreasing the weight of of the smaller ones, which also
affects size-frequency statistics. Additionally, it is likely that thousands of smaller landslides could not
be mapped from the medium-resolution Sentinel-2 imagery (10 m) and the higher resolution imagery
(0.5 – 1 m) available on GEPro for 50% of the target area before the end of 2021. To refine our landslide
mapping in future, higher resolution imagery must be used for the whole area affected by the 2021 event,
and automatic landslide identification techniques shall be applied by combining image analysis and
machine learning as proposed by Amatya et al. (2021). Actually, the 'manual' mapping applied now
would take too much time to outline the many thousands of very small landslides that have not been
identified so far. Those would contribute to the increase of the weight of the smaller landslides in the
2021 inventory, especially of those smaller than 2000 m$^2$.
From the preceding we can infer that the mapping of additional smaller landslides will not really modify
the power-law decay part presented in Fig. 7 (for which also the 2021 catalogue can be considered as
complete), and increase the absolute value of the related decay exponent – which is smaller (2.0-2.1)
than the one proposed by Malamud et al. (2004) as 'universal' value (2.4) for landslide events.
Interestingly, the same graph in Fig. 7 shows that the exponent of the power-law decay part of the 2010
landslide frequency-density distribution is even higher (2.6) than the value proposed by Malamud et al.
(2004). One reason for the different values could be that the decay exponent value of -2.4 is simply not
as universal as suggested by Malamud et al. (2004), a hypothesis that is supported by the findings of Van
den Eeckhaut et al. (2007), Stark and Guzzetti (2009) and Tanyas et al. (2019) who also reported varying
exponent values (that are still close to the one of -2.4, typically between -1.8 and -2.8). Especially, the
higher value obtained for the Harp et al. (2016) inventory indicates that this value could be influenced
by the mapping technique (also related to the availability of high-resolution imagery, and to the outlining
of coherent landslides vs distinguishing between individual landslide zones within a larger mass
movement).
Another reason for the smaller number of landslides mapped for the 2021 event (that does not exclude
the first one) would be related to the fact that the landslides triggered in 2010 mainly consisted of narrow
slides and flows in weathered limestone rocks. Thus, the type of rock affected could have an influence

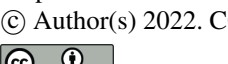


on the power-law decay of landslide size-frequency distributions. For instance, Havenith et al. (2015)

showed for the Tien Shan Mountains that the landslide distribution in soft soils are marked by a higher

decay exponent (~2.1) than the one in hard rocks (exponent of -1.9 for mass movements in areas mainly

made of magmatic or sedimentary hard rocks). Anyway, in order to be able to exclude any artificial effect

on the size-frequency statistics, the new 2021 landslide event catalogue has to be revised once all the

area is covered by higher resolution imagery (at least <= 1 m pixel size). After this work, we may then

also provide a more definite answer to the related questions: first, is the new 'complete' inventory for

2021 characterized by the same power-law decay exponent as the present one (= - 2) or do we obtain a

new one closer to the value (= -2.4) proposed by Malamud et al. (2004)? Second, shall we observe the

rollover for a landslide class of about 300 m$^2$, or for a different class, or for none (at least not above 100

m$^2$), just as it seems to be the case for the 2010 landslide inventory by Harp et al. (2016)? If all parameter

values will be close to those predicted by the equations by Malamud et al. (2004), then, the future 2021

inventory should contain close to 100,000 landslide polygons, according to the graph shown in Fig. 7 –

which seems to be an unrealistically high number, even if that inventory will be refined as proposed

above. If, however, a rollover is observed for a landslide class of about 300 m$^2$ (now it appears for a

much larger class of about 3000 m$^2$) as proposed by Malamud et al. (2004), but the power-law decay is

marked by a lower absolute value of the exponent (closer to present one, near -2.1), then, the new

inventory should contain about 20,000 landslides (closer to the number predicted by Havenith et al., 2016

for such an event) according to the graphical prediction in Fig. 7. Certainly, there is also the possibility

that none of the predictions proposed by Malamud et al. (2004) would be verified after landslide

remapping on the basis of higher resolution imagery and applying automatic landslide detection

techniques – and then no estimate of the number of landslides contained in the future inventory can be

provided right now. Among the three possibilities, we think that option 2, combining a rollover for a

landslide surface area of 300 m$^2$ with a lower absolute power-law decay component (~2.1) shall be the

most realistic one marking a complete 2021 landslide inventory. This assessment is based on our present

mapping experience. Also, the total number of landslides of 20,000, predicted by the frequency-density

curves proposed by Malamud et al. (2004) for those values (see graph in Fig. 7), would be of the same

order of magnitude as the number predicted by Havenith et al. (2016) for such a Mw=7.2 earthquake and

the observed background conditions (~13,500).

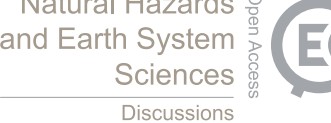

Disregarding the likely influence of the inventory completeness on the different size-frequency statistics
of the two landslides inventories related to the 2010 and 2021 events, the changing climatic conditions
could also affect those statistics. Notably, it could be shown that the climatic contribution to landslide
activity in 2021 might be twofold: first, some post-seismic intensification of slope failures could be
observed after the tropical storm Grace event that had crossed the earthquake region on August 16-17,
two days after the main shock. However, related effects cannot really be quantified as only 10% of the
total surface area potentially affected by the earthquake appeared as cloud-free on imagery available right
after the August 14 main shock and before August 16 (Grace event). For those limited areas, we estimate
that storm Grace caused additional 10-15 % of slope failures with respect to the purely earthquake-
induced landslide activation. Second, by comparing the 2016-2017 landslide distribution with the one
observed after August 14, 2021, it can be seen that most of the October 2016 – end 2017 landslides
occurred within the same region as the 2021 ones and most were clearly reactivated by the seismic
shaking in August 2021 (while also many of them had been revegetated in between). Above we could
show that Hurricane Matthew had crossed the western part of the peninsula in October 2016, producing
an abnormal amount of precipitation precisely over the area recently hit by the earthquake (see GPM
maps in Fig. 8), and where also numerous landslides had occurred just after mid-October 2016. Therefore,
it is very likely that this climatic event has triggered many of the pre-seismic (pre-2021 and even pre-
2018) landslides, which preconditioned slope instability all over the area hit by the 2021 earthquake. The
double hurricane effect (by Matthew in 2016 and by Grace just after the 2021 main shock) observed in
the area hit by 2021 earthquake could be responsible for the proportionally larger size of the 2021
landslides (preconditioning of slope failures and post-seismic intensification) compared to the 2010 ones,
and thus explain the lower absolute value of the power-law decay. As indicated above for other aspects,
the related conclusion still requires some remapping of the 2021 landslides and 'finetuning' of their
outlines.
Additionally, the combined seismic and climatic influence could also explain the very different spatial
landslide distribution characteristics of the 2010 and 2021 catalogues: the relative dispersion of
landslides observed after the 2010 event could thus be partly related to the spatially highly variable effect
of tropical storms and hurricanes on landslide activity (acting over a longer period, with an influence that
could last over tens of years), partly overprinting the more concentrated seismic effect (resulting in



clusters of mass movements near the seismic source zone). The same dispersion might also have been
observed for the 2021 event if the central part of the seismically affected area had not been hit by that
major climatic event just five years before – doubling the landslide concentration effect in that area
(specifically for this event). However, we acknowledge that a quantification of these opposite effects of
climatic events, both on landslide dispersion and on their concentration, requires a more detailed analysis,
also focusing on specific sites, by completing numerical simulations of mass movements affected by
variable climatic (modelling changing groundwater level) and seismic influences (including the effect of
rock structures and types of lithologies and morphologies on shaking polarization and amplification). A
related landslide spatial distribution analysis should then also consider the influence of extensive
deforestation on slope destabilization, all over the country of Haiti. Actually, deforestation is responsible
for the decrease of 90% of the primary forest over the last few tens of years, especially in the southern
regions of Haiti where the two earthquake events had occurred (see Hedges et al., 2018). As mostly
shallow landslides occurred in 2010 and 2021, the effect of deforestation on the destabilization of shallow
soils and weathered rock cover must be taken into consideration for landslide occurrence prediction.
Such an extensive study would thus require the creation of an integrated seismotectonic-morpho-
geological-climatic-soil cover model allowing us to fully understand changing landslide activity in Haiti
– which is not the target of the present analysis.
As for the climatic part, here, we only present regional data to outline some general seismic influences
on landslide activity induced by the 2010 and 2021 earthquakes. Related maps (Fig. 9) show that the
aforementioned gap of landslides between the areas affected by the earthquakes in 2010 and 2021 would
indeed be marked both by a lower shaking intensity (but showing values that are still larger than the
threshold Ia values observed elsewhere for landslide occurrence) and lower landslide susceptibility (still
to be published).
**5    Conclusions**
In this paper we first presented the new landslide inventory created for the Mw=7.2 Nippes earthquake
that occurred on August 14, 2021, in Haiti. Related spatial and statistical characteristics have been
compared with those of the landslides mapped by others for the previous, Mw=7.0, January 12 (2010),
earthquake that had occurred along the same fault zone (EPGF zone) but more to the East. Considering
a series of uncertainties affecting the landslide statistics (related to the mapping technique, including the
uncertain number of particularly small landslides triggered in 2021) and the environmental information
(including the climatic and geological conditions), this comparison allowed us to highlight the following
points: 1) the 2021 earthquake triggered landslides over wider surface areas than the one in 2010; 2) size-
frequency statistics computed for the two landslide catalogues present a clear power-law decay, marked
by different exponent values; however, a rollover is only observed for the 2021 inventory (but for a
relatively large landslide area class, hinting at an incompleteness of the inventory, for the smaller
landslides); for the 2010 data such feature does not appear, at least not for landslide sizes above 100 m$^2$,
hinting at an 'over-completeness' of that inventory (compared with others published); 3) climatic
preconditioning of slope instability could be 'proved' for the 2021 event, mainly in connection with the
impacts of recent hurricanes in the 2021 affected region, which could also partly explain the more
extensive landslide activity observed in 2021; 4) the 2010 landslides seem to be more dispersed around
the epicentral area than the 2021 slope failures, which could be due to the opposite climatic effect
inducing spatially more variable slope destabilization (also as no particular storm had hit the 2010
affected region just before or after the seismic event, as it was the case in 2021); this dispersion effect
can also be enhanced by the spatially varying deforestation that is locally very intense in the target areas.
We estimate that this proof of a combined seismic and climatic influence on landslide activity (possibly
augmented by morpho-geological and soil cover effects not studied in detail here) opens new avenues
for geohazard research, especially for regions like Haiti that are regularly hit both by severe earthquakes
and weather events. We also think that preconditioning of slope failures by multiple events over longer
terms, including by former earthquakes, should be studied more in detail as this preconditioning could
highly contribute both to regional and local landslide hazards over short and longer terms. A full analysis
of such a scenario would require the development of an integrated seismotectonic-morpho-geological-
climatic-soil (and vegetation) cover model, which can only be completed through an extensive
international multi-disciplinary collaboration around this target – which is obviously missing for Haiti.
Assessment of related risk would further require the involvement of experts in social geography and
economy. Only when this goal is achieved, we could really work on the prevention of at least parts of
another future earthquake disaster in Haiti.




**Acknowledgments**
This study was partly supported by the 'Earthquake Hazard and Vulnerability assessment – developing
innovative solutions for sustainable Risk Reduction and Communication in Haiti' project funding (2019-
2024) provided by the Belgian ARES – ACADÉMIE DE RECHERCHE ET D'ENSEIGNEMENT
SUPÉRIEUR.

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

**Annex**

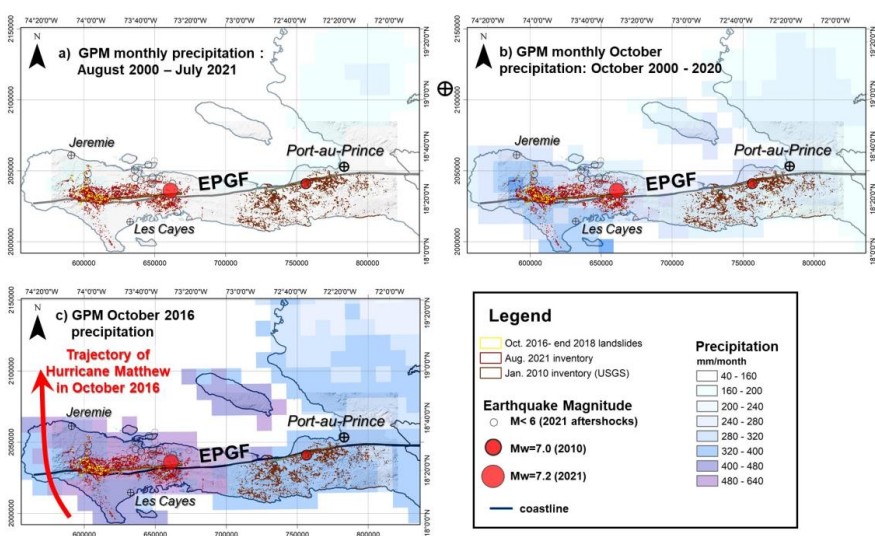

**Figure A1: Monthly Global Precipitation Measurement Mission (NASA) maps (0.1° resolution, values in**
**mm/month) for southwestern Haiti, (a) for all months between August 2000 and July 2021, (b) for the month**
**of October between 2000 and 2020, and (c) for October 2016 (marked by the Hurricane Matthew event).**