# Peer review of "Earthquake-induced landslides in Haiti: seismotectonic"

_Natural Hazards and Earth System Sciences, 2022_

## Referee Comment (RC1)

Review of Balder-Havenith et al.,

The paper by Havenith and collaborators is presenting various data and analysis about recent landsliding in Haiti, caused by the 2021 earthquake and in parts by the 2016 and 2021 Hurricanes, and putting that in to context by comparing that to coseismic landslide inventories from the 2010 earthquake and to some literature model.
However, the paper is difficult to read because of its poor structure (no problem with the language overall). Indeed the paper feels more like a technical report : The introduction do not present any clear scientific question, and do not give the state of the art knowledge on such questions, and many bits of data/methods are in the intro or arrive in the middle of the result, and a lot of the discussion is lengthy and speculative with very limited usage of the existing literature.

Clearly the introduction should be completely rewritten to explain what knowledge gap about EQ induced landslides the author plan to address and to include a minimum of references about these different questions (several useful references are already in the discussion, many are missing, I have given a few).

Beyond that there are multiple problems with the results.
For example, basically the author say almost directly that they could not map small landslides AND have amalgamated landslide into single polygon during the mapping, but then they insist on doing frequency size statistics and landslide number analysis, whereas multiple work in the literature conclude that this is poor practice and will give biased results (and thus should just be avoided). Given that a large part of the results and discussion of the paper is about that, this is a major concern.

**Additionally I read on the NHESS guidelines :  Localised case studies with no broader implications ARE OUT-OF-SCOPE of the journal …**

After reading the whole paper, in its current state it is not so clear what are the broader implications and it indeed mostly looks like a localized study about a recent landslide event, but with a very weak novelty :
1) landslide statistics and number are biased, and thus not really useful. In the conclusion the authors even put forward L920 what appears as a wrong conclusion (based on an incorrect analysis ?) of the 2010 event, in contradiction with the literature (see final Line by Line comments)
2) Post seismic triggering/ reactivation from Hurricane Grace could be an interesting topic but it is very poorly constrained (10% area and only under the very poor description of "intensification"; see minor comments).
3) Total landslide area, affected area and link with Arias intensity : well there is some comparison with the past EQ and previous model but without a general framework nor a discussion accounting with more recent models / studies.
4) Preconditioning from Matthew : Well apart from saying that there were new landslides caused by Matthew (which did caused a lot of rainfall in October 2016) the authors do not propose any explanation in terms of how this would affect the 2021 coseismic landsliding

So clearly there is a lot of work for this paper to become clearer, insist on more robust results and highlight what is innovative and broadly useable. If this work cannot be done my impressions is that this work should rather go into a specialist journal, and be trimmed down to mainly present the new data and not much more.

**Line By Line Comments**

Fig1 : Lot of problems to fix… The hazard map is of very little use. The map with landslides should be the whole figure (enlarge it a lot ! ) then the legend box can go inside, in the upper left corner (mainly sea…) The map should be zoomed, and the full Hispaniola island could just be a mini inset.
Note that all figures could become larger and more visible by making the legend box an inset …
Make more difference in color between 2010 and 2021 landslides. Probably you need 5 colors :
2010 slides / 2016-2018 slides / 2021 EQ only / 2021 Grace and EQ … For sure you need the footprint of 1) High res imagery ;  and 2) as much as possible the area with cloud free imagery (GEPro or S2) showing the immediate coseismic EQ…

L127: rather say  " local members of our research team…"

L158 and L165 : Authors decided to amalgamate neighboring landslides. As a results size statistics (including on the power law decay exponent, will be biased see Marc and Hovius 2015) and the point of comparing it to other catalogue is not really interesting…
 Only the total area is useful and probably the spatial pattern, why don't the authors focus on these points instead ?

L203 : Widening of 10-15% due to Grace : Please be very clear here, do your mean 10-15% more area due to exclusively reactivation ? (that's what "widening" would imply to me but it is contradicted when just after you say increase in total area *and numbers* ! )
or 10-15% more area due to both reactivation/enlargement AND new slides ! That's different. And why can't you discriminate (where you have the imagery you should be able to) ?
Grace could certainly have triggered new landslides (and all the work on "post-seismic landsliding" suggests that rainfall event directly following large EQ produce even more NEW landslides that pre EQ rainfall events … )
And importantly before extrapolating you should state in how much % of the area affected you could be sure what you mapped is coseismic, and how much is mixing co and post-seismic … Because if it is small it just makes no sense to take a bulk value …

Figure 4: commenting on the landslides designed by the 6 arrows from left to right… Of course I cannnot zoom as much as you but my interpretation based on the pre/post image would be :
 1$^{st}$ arrow: New ; 2$^{nd}$ : new ; 3$^{rd}$ : reactivation ; 4$^{th}$ : New ; 5$^{th}$ : reactivation ; 6$^{th}$ : New (or reactivation of a small one? Hard to see)
So "intensification" of denudation is a very poor description. You need to classify the different process: reactivation when the Grace polygon intersect or contain a coseismic polygon, new when it does not.

L230-237 : As described now, nobody could use Eq 1 without a full reading of Havenith 2016 (which by the way is NOT in the references list so I am not even sure if I should read Havenith et al 2006 or 2015 … or if the ref is just missing … ) . You need to say what is the value of the different param and/or based on which data you have derived them … ! SO basically put Table 2 here and explain it !!

Also there exist other models to predict area for example with some discussion suggesting that number is a quite problematic variable to predict (because it is ill-defined , depending on mapping philosophy, imagery resolution, mapping errors etc etc) … Area in contrast is much more robust. See discussion in Marc et al 2016.

L242 : What is the area "potentially affected by landslides" ?
It seems to me like an ill-defined concept (because it is essentially not possible to define it from observation in the field or by remote-sensing … Why not to use the concept of area of distribution of landslides (see Marc et al 2017, Tanyas and Lombardi 2019) which can be measured, and has been discussed ?
If what you refer to is the same thing you should still cite and discuss these more recent approaches and not limit yourself to comparing your model to one derived in 1989 !!
Note one key point maybe whether or not it is meaningfull to make a convex hull with ALL landslides or only with a significative fraction (say the smallest surface containing 95% of area which would be robust to isolated landslide, mapping errors or uncertainties etc see Marc et al 2017 for discussion). This may remove the few landslides at the end of the Southern Peninsula and change quite a bit your potential area…

L284 : As far as I understand you should not say that you "compute" Arias intensity but directly say that you "model" it, because you do not use seismic record but an empirical equation…

L374 : "according to our estimate" based on what ? Field ? Imagery ? Geometric assumptions ? Please specify …

Section 3.2 is mostly useless… The authors discuss mapping artifacts which are known and have been presented and discussed elsewhere in the literature… Landslide Numbers are a misleading metrics because it is hugely dependent on imagery resolution, amalgamation etc … See Marc and Hovius 2015, Marc et al 2016.

L417-418 : And there are good reason to do that : because failure are likely individual while coalescence of the deposit is just a contingency. Independent mapping allows to estimate volume with some confidence, as well as statistics (frequency size, slope gradient at source location  etc)...

L430-440 : These parameters should be placed in the methods …

L480 : Why do you suddenly focus on landslide average and median area ? For a power-law distribution it does not make much sense (and it is hugely sensitive again to mapping techniques and amalgamation … )

L482 : An example of fallacious (or simply lazy) logic: "confirming on the one hand that larger landslides were triggered in 2021, and, on the other hand, that many more small landslides were mapped by Harp et al. (2016) than by our team for 2021"
The median is different, and you cannot say that it is "confirming" A (triggering larger) and B (mapping bias). Because both A OR B  are enough to change the median … So as of now the mapping bias that you recognize mean that you cannot really conclude whether the triggering properties had an effect on the average/median size … For a discussion on effects of EQ shaking on landslide size (based on power law decay variations see Marc et al 2016 and Valagussa et al 2019

L594 : Careful here : you state "theoretical FD curves" but Malamud 2004 never say they were "theoretical" and has no theory to explain their emergence. They state in their paper that they are empirical distribution that are better fitting a number of distribution …

So you seem to greatly over-estimate the Malamud 2004 paper, which indeed was important in showing that the rool-over and powerlaw decay should be considered, but did not close the topic at all...

There has been much more work since on the (potential) physical reason of this emergence (see Frattini and Crosta 2013, Stark and Guzzetti 2009, Jeandet et al 2019) and on whether or not alternative (ie log-normal fit to the FD curves) could also work (see Marc et al 2018, Last paper ENglish)

L647-651 : This could go to methods / Data...

L655 : This sentence does not help much… you can remove it safely .

L660-670 : I do not understand why to do a monthly comparison ?? clearly the timescale of rainfall accumulation is much shorter, Grace maybe delivered most of its rainfall in a few days. So it would seem much more logical to look at the peak rainfall over 3 days (for example) and compute over 20 years the max total rainfall over 3 consecutive days in all past years (this should also capture Matthew for example… and better allow to compare various hurricanes… )

L686 : What is that ?? "with the average precipitation of all months of October between 2000 and 2020, of 200-320 mm," How can you give a range when talking about an average ?? do you mean 260 +/-60 (mean and 1 standard deviation) ? That is a reasonable way to express results but you cannot leave the sentence as it is now …

L699 : What do you mean by that ? Your observations allow you to conclude that Matthew triggered landslides in the same zone, but then what ? Do you think it depleted the "stock" of hillslope near to failure (and thus should reduce the number of coseismic landslides) ? Or do you think it made hillslopes even more prone to failure ? How ?
Clearly not by reactivation as I do not see how 324 failure could have had a major impact on the following >7000 failures …
It is not very clear to what the authors are hinting.

L 707 : "notably to explain why numerous landslides had occurred very far from the seismic source zone." What do you mean by numerous ? When I look at your figures I see a vast majority of landslides within 20 km of the fault ….

Fig 8 : The map a is pretty much useless as there is a dry and wet season, the average monthly rainfall is not interesting … Even in the text you do not make this mistake and compare the Huricane to the long term average of its month of occurrence (during the Hurricane season). So I would rather suggest that you show only Two maps, normalized by the long term mean of the relevant month … But even better would be to show to maps of the total rainfall during the weeks ( or pentads/triads) of the Hurricane, divided by the average rainfall of the 20 wettest weeks (pentads/triads) of each year … See my comment above about the time scale of

Section 3.4
You did not explain Why you prefer to use a first order model (Keefer and Wilson 1989) rather than a USGS Shakemap ?
I see argument in favor of both approach but there should be some line about that in the methods…

L762-775 : The discussion on the deep seated landslides is not very convincing … The authors state that the lack of large landslide volume may be due to the lack of high relief terrain… But there are counterexamples : For example the largest landslides of the Mw 7.9 Gorkha earthquake in the Nepal Himalayas are of a few Mm3 … (see Lacroix 2016).

Mw 7.9 Denali or Mw 7.8 Kaikoura would also be on the edge of this criteria (as they both have only their largest landslides about 20 Mm3 see Jibson et al 2006 ; Massey et al 2018 , and then smaller landslides of few Mm3).

These three cases have all large relief…

Also why to choose this 20 M m3 limit ?

L793 – 802 : This paragraph is basically explaining why the current inventory cannot be useful to understand size statistics or landslide numbers … For me if your data is not useful for a scientific problem you better not make your paper around it …

The author could state for example that for the moment their mapping allows to constrain the total landslide area, and the affected area, and the question of EQ/rainfall conjunction in triggering landslide and focus on these topics. They could then state that without HR imagery and a mapping focused at avoiding amalgamation landslide number and size statistics are not pursued for the moment.

L810-813 : These statement are unsupported and very possibly wrong: it has been shown (based on several landslide inventories) that amalgamation is changing the power law exponent (Marc and Hovius 2015), so it is not only the fact that more small landslides could be mapped but also the fact that to remove mapping bias, you would need to split large amalgamated polygon into multiple smaller landslides…

L817 : I do not see the point of this paragraph : When you have all successive paper that contradict one paper (here the statement of Malamud 2004 that 2.4 may be a universal decay exponent) it seems poor scientific method to start saying that you believe in a paper, then say that your data contradicts it and that you hypothesize that in the end this paper may be wrong and then conclude in discussion that, actually, all the paper that you did not mention in the intro are supporting your conclusions ….

L 856 : "Disregarding the likely influence of the inventory completeness on the different size-frequency statistics"

You should not … We don't need so much speculation about an effect that is relatively obvious and well identified in the literature (and an effect that you CAN correct, even if not at present )….

L918-922: This opposed to what you can see in any other paper (see Fig below from  Tanyas et al 2019)  that there is a clear roll over just below 100 m2 … Why did you not look at it ?

[Figure]

Further on what is based your conclusions on "over-completenes" ? If you look at another figure from Tanyas et al., 2019 again a figure clearly show something like 5-6 inventories with Roll over below 100 m2 … (Of course it could not be true for older catalogues (as at least 2 of the ones used by Malamud 2004) mapped ONLY with lower resolution that could not resolve such small size). It just seems that in some landscape the rollover can be at small scale, and yes you should not believe Malamud and its "completeness" scale that is assuming the same roll over and power law decay for all event...

[Figure]

**References**

Frattini, P. and Crosta, G. B.: The role of material properties and landscape morphology on landslide size distributions, 361, 310–319, https://doi.org/10.1016/j.epsl.2012.10.029, 2013.

Jeandet, L., Steer, P., Lague, D., and Davy, P.: Coulomb Mechanics and Relief Constraints Explain Landslide Size Distribution, 46, 4258–4266, https://doi.org/10.1029/2019GL082351, 2019.
Lacroix, P.: Landslides triggered by the Gorkha earthquake in the Langtang valley, volumes and initiation processes, 68, 1–10, https://doi.org/10.1186/s40623-016-0423-3, 2016.

Massey, C., Townsend, D., Rathje, E., Allstadt, K. E., Lukovic, B., Kaneko, Y., Bradley, B., Wartman, J., Jibson, R. W., Petley, D. N., Horspool, N., Hamling, I., Carey, J., Cox, S., Davidson, J., Dellow, S., Godt, J. W., Holden, C., Jones, K., Kaiser, A., Little, M., Lyndsell, B., McColl, S., Morgenstern, R., Rengers, F. K., Rhoades, D., Rosser, B., Strong, D., Singeisen, C., and Villeneuve, M.: Landslides Triggered by the 14 November 2016 Mw 7.8 Kaikoura Earthquake, New ZealandLandslides Triggered by the 14 November 2016 Mw 7.8 Kaikoura Earthquake, New Zealand, https://doi.org/10.1785/0120170305, 2018.
Marc, O. and Hovius, N.: Amalgamation in landslide maps: effects and automatic detection, 15, 723–733, https://doi.org/10.5194/nhess-15-723-2015, 2015.

Marc, O., Hovius, N., Meunier, P., Gorum, T., and Uchida, T.: A seismologically consistent expression for the total area and volume of earthquake-triggered landsliding, 121, 640–663, https://doi.org/10.1002/2015JF003732, 2016.
Marc, O., Meunier, P., and Hovius, N.: Prediction of the area affected by earthquake-induced landsliding based on seismological parameters, 17, 1159–1175, https://doi.org/10.5194/nhess-17-1159-2017, 2017.
Marc, O., Stumpf, A., Malet, J.-P., Gosset, M., Uchida, T., and Chiang, S.-H.: Initial insights from a global database of rainfall-induced landslide inventories: the weak influence of slope and strong influence of total storm rainfall, 6, 903–922, https://doi.org/10.5194/esurf-6-903-2018, 2018.
Tanyaş, H. and Lombardo, L.: Variation in landslide-affected area under the control of ground motion and topography, 260, 105229, https://doi.org/10.1016/j.enggeo.2019.105229, 2019.
Tanyaş, H., Westen, C. J. van, Allstadt, K. E., and Jibson, R. W.: Factors controlling landslide frequency–area distributions, 44, 900–917, https://doi.org/10.1002/esp.4543, 2019.

Valagussa, A., Marc, O., Frattini, P., and Crosta, G. B.: Seismic and geological controls on earthquake-induced landslide size, 506, 268–281, https://doi.org/10.1016/j.epsl.2018.11.005, 2019.

Jibson et al., 2006

    R.W. Jibson, E.L. Harp, W. Schulz, D.K. Keefer

    **Large rock avalanches triggered by the M 7.9 Denali Fault, Alaska, earthquake of 3 November 2002**

    Eng. Geol., 83 (2006), pp. 144-160

---

## Referee Comment (RC2)

[referee-annotated manuscript omitted]